# Learning earthquake ground motions via conditional generative modeling

Pu Ren [1] ✉, Rie Nakata [2,3,4], Maxime Lacour [4,5], Ilan Naiman [6], Nori Nakata [2,4,7], Jialin Song [4,8], Zhengfa Bi [2], Osman Asif Malik [9], Dmitriy Morozov [9], Omri Azencot [4,6], N. Benjamin Erichson [1,4] & Michael W. Mahoney [1,4,10]

Predicting high-fidelity ground motions for future earthquakes is crucial for seismic hazard assessment and infrastructure resilience. Conventional empirical simulations suffer from sparse sensor distribution and geographically localized earthquake locations, while physics-based methods are computationally intensive and require accurate representations of Earth structures and earthquake sources. We propose an artificial intelligence (AI) spectrogram generator, Conditional Generative Modeling for Ground Motion (CGM-GM). CGM-GM leverages earthquake magnitudes and geographic coordinates of earthquakes and sensors as inputs, when postprocessed with phase information, capturing spatially continuous Fourier amplitude spectra (FAS) as well as properties such as P and S arrivals, and waveform durations, without explicit physics constraints. This is achieved through a probabilistic autoencoder that extracts latent distributions in the time-frequency domain and variational sequential models for prior and posterior distributions. We evaluate the performance of CGM-GM using small-magnitude earthquake records from the San Francisco Bay Area, a region with high seismic risks. Here, we report that CGM-GM demonstrates potential for complementing physics-based simulations and non-ergodic empirical ground motion models, as well as shows promise in seismology and beyond.

The accurate prediction of ground motion waveforms and their characteristics for future earthquakes is crucial for assessing seismic hazards and ensuring the safety and resilience of critical infrastructure. However, it is challenging and resource-intensive to obtain comprehensive ground motion observation across a wide geographic area. Furthermore, predictions of earthquake rupture processes and estimates of the Earth's elastic model remain to exhibit significant uncertainties. The development of precise and robust ground motion prediction methodologies has long been of great interest in seismology and earthquake engineering to complement the limited recorded data.

Existing ground motion simulation studies branch into two streams: stochastic and physics-based approaches. The first stream, stochastic methods, is rooted in the modulation of Gaussian white noise to reproduce the desired ground motion characteristics[1-4]. They provide a computationally efficient framework to synthesize

[1]Scientific Data Division, Lawrence Berkeley National Laboratory, Berkeley, CA, USA. [2]Energy Geosciences Division, Lawrence Berkeley National Laboratory, Berkeley, CA, USA. [3]Earthquake Research Institute, University of Tokyo, Tokyo, Japan. [4]International Computer Science Institute, Berkeley, CA, USA. [5]Department of Civil and Environmental Engineering, University of California, Berkeley, CA, USA. [6]Department of Computer Science, Ben Gurion University of the Negev, Beer-Sheva, Israel. [7]Department of Earth, Atmospheric and Planetary Sciences, Massachusetts Institute of Technology, Cambridge, MA, USA. [8]School of Computing Science, Simon Fraser University, Burnaby, BC, Canada. [9]Applied Mathematics and Computational Research Division, Lawrence Berkeley National Laboratory, Berkeley, CA, USA. [10]Department of Statistics, University of California, Berkeley, CA, USA. ✉e-mail: ren.pu@northeastern.edu

ground motion data by calibrating stochastic process-based models to match the historical recordings. However, potential limitations exist in representing spatial continuity and physical phenomena. The second stream, physics-based methods, is based on the numerical solution of wave equations[5–9] while considering comprehensive physical characteristics, including fault ruptures[10–13], heterogeneous earth media, and site-specific effects. Although recent advancements in high-performance computing enable simulating high-frequency waveforms of large-magnitude earthquakes (e.g., up to 10 Hz)[14], physics-based methods are computationally demanding. For example, ground motion simulations of the San Francisco Area over a domain of $120 \times 90 \times 35$ km$^3$ require 128 NVIDIA A100 GPU nodes and take 6 h to compute up to a frequency of 5 Hz. Simulations at higher frequencies tend to be computationally prohibitive and these data are typically complemented by stochastic simulations[15,16]. Furthermore, physics-based simulations face challenges from significant uncertainties in wave theory, subsurface elastic models, and source characteristics.

More recently, machine learning (ML) and artificial intelligence (AI) have shed new light on this classic task, primarily through their capability of accelerating earthquake modeling processes. One representative line of work involves the use of neural operators for modeling seismic wave propagation[17,18] and extending their application to complex 3D elastic wave phenomena[19,20]. Although these data-driven ML methods show efficiency by avoiding the stringent time-step constraints in traditional time-domain physics-based numerical approaches, they require a large amount of high-fidelity data. Researchers have also resorted to incorporating physical constraints into ML models, such as physics-informed neural networks (PINNs)[21]. In particular, by leveraging physical principles as a prior, PINNs have been used for predicting ground motions with a limited amount of training data[22–24]. However, PINNs are known to exhibit fundamental failure modes in network optimization[25–27]. Moreover, due to their specific design of objective functions, PINNs encounter significant limitations in generalizing to different initial conditions and subsurface elastic models. PINNs can be regarded as physics-based and still suffer from uncertainties in the problem setup. In addition, the spectral bias of fully-connected neural networks used by both neural operators and PINNs typically constrains the resolution to low frequencies[23], making broadband synthesis of waveforms challenging. Furthermore, graph neural networks (GNNs) have also been employed to earthquake ground motion synthesis[28–30], leveraging their ability to capture spatial correlations. These models are capable of predicting ground motion intensities at more distant stations within the seismic network.

Generative modeling has emerged as an alternative approach for scientific modeling to capture the complexities of natural phenomena, as demonstrated for fluid dynamics[31,32] and molecular science[33,34]. It is an inherently stochastic method that incorporates random noise as inputs and uses probabilistic processes to generate diverse and realistic data. A detailed discussion of commonly used generative models can be found in Supplementary Section 1.2. In the context of earthquake ground motions, generative modeling can produce various waveforms while capturing model uncertainties. This capability is critical due to the significant variability and unpredictability of real-world earthquake ground motions, influenced by factors such as seismic sources, propagation paths, and site conditions. Generative models, which are not governed by specific wave equations or subsurface models, need to learn meaningful wave physics (i.e., governing equations) and site/source conditions from sparse and irregular sensor and earthquake distributions. Generative Adversarial Networks (GANs) and their variants[35–39] have demonstrated success in simulating ground motions, which paves the way for generative modeling of ground motions. Their ability to produce broadband waveforms makes them a valuable tool in generative modeling. However, their training process can be sensitive to design choices, often requiring careful tuning to mitigate challenges such as mode collapse and instability[40]. These prior works focus on using distances, magnitudes, and near-surface velocity structures ($Vs_{30}$) as inputs, allowing for capturing ergodic stochastic characteristics. Nevertheless, to date, generative AI models have not been demonstrated to capture spatially variable wave propagation effects.

In this paper, we demonstrate that generative modeling can be more robust and powerful by introducing Variational Autoencoder (VAE) and incorporating geospatial coordinate information. This approach presents spatial continuity in median spectral components of the generated ground motions. To achieve this goal, we introduce a Conditional Generative modeling framework for Ground Motion (CGM-GM). Recent work[41] indicates that dynamic VAE models are a flexible, user-friendly, and robust alternative to GAN models for time series generation, mitigating common issues such as mode collapse and training instability. More broadly, this feature enhances the accessibility of this framework for seismologists who may not have extensive experience with generative modeling methods. The primary advantage of dynamic VAE models is their use of variational sequential architectures (e.g., recurrent neural networks) in both prior and posterior distributions, which facilitates the capture of the temporal evolution (dynamics) of time series. Our main methodological contribution lies in the design of dynamic VAE models for learning time-frequency information and the conditional embedding of physical parameters, including earthquake magnitudes, depths, and geospatial coordinates. This strategy enables the CGM-GM framework to implicitly learn the underlying physics and spatial heterogeneity from observation data, even without explicitly incorporating physical principles. As a result, our generative models can produce realistic simulations that are aware of as a combination of source, path, and site effects.

We demonstrate the effectiveness of our method by focusing on the San Francisco Bay Area (SFBA). Despite the relative infrequency of large-magnitude earthquakes, the densely populated SFBA remains a region of significant public interest due to its history of such events. This is different from previous studies[35,36,38] that have leveraged a much larger number of large-magnitude earthquakes recorded in Japan. Instead, our work uses small-magnitude earthquakes that are crucial for characterizing earthquake ground motion, especially for the linear effects of the path and site. Studying small-magnitude earthquakes allows us to estimate the region-specific variations of the ground motion, which can help reduce seismic hazard in areas where no large earthquake records are available[42,43]. This task presents a significant challenge since our model needs to generalize well with lower signal-to-noise ratio (S/N) data. Moreover, the seismic data are recorded at a limited number of stations per event due to the released seismic energy and attenuation, which leads to a relatively sparse seismic network compared to wavelengths.

Here, we show the performance of our proposed model in producing Fourier Amplitude Spectra (FAS) maps and their spatial heterogeneity as well as reproducing properties such as P and S arrivals, and waveform durations. These features distinguish our framework from existing generative models in the context of ground motion generation. Additionally, CGM-GM results exhibit good agreement between generated samples and ground truth data across the entire frequency range ([2, 15] Hz), including waveform shapes, peak ground velocity (PGV) distributions, FAS, and arrival time of earthquake ground motions in the SFBA. We recognize that specifying only magnitude and source location does not uniquely define an earthquake event, and limits the capability to generate a set of waveforms from a single event. Nevertheless, we demonstrate that the model is capable of capturing spatial continuity in the median-generated FAS.

## Results

### Datasets

The dataset for the study of small-magnitude earthquakes in 1990–2022 in the San Francisco region is downloaded from the Northern California Earthquake Data Center (NCEDC) database. The stations of interest are chosen within a 50 km radius from the Hayward fault, and the events with magnitude $M < 4$ recorded within a rupture distance $R_{rup} < 100$ km from the selected stations are included. We focus on two horizontal components H1 and H2 of particle velocity, considering their direct importance for earthquake hazard analysis. The H1 and H2 components correspond to the East-West (E-W) and North-South (N-S) directions, respectively. After the data selection process illustrated in the Methods section, we retain 5, 108 and 5, 301 recordings available for H1 and H2 components, all within the frequency range of [2,15] Hz.

### Conditional generative model

We present our CGM-GM framework based on a conditional dynamic VAE framework, as shown in Fig. 1a–d for generating realistic ground-motion time-series data. The objective is to develop a generative model for predicting ground motions $\mathcal{D}^*$ at unobserved sources and site locations, with varying earthquake magnitudes, based on actual sparse seismic recordings $\mathcal{D}$. Firstly, in our model, we use Short-Time Fourier Transform (STFT)[44,45] to decompose the time sequence data $\mathbf{x}_{1:T}$ into amplitude and phase information in each time window, where $T$ is the length of the ground motion time series. STFT is an effective technique to extract the time and frequency information, which is also considered in the previous implementation of the GAN model for ground motion generation[36]. Our dynamic VAE model is trained on amplitude information $\mathbf{A}_{1:\tau}$, which is in the time-frequency domain with a new time sample $\tau$ used in STFT ($\tau \le T$). The key methodological contribution lies in the specific design of the prior and posterior distributions in VAE models, where we incorporate temporal dynamics using recurrent neural networks (RNNs) for learning time-frequency information[41,46]. Next, we employ two strategies to obtain phase information and apply inverse STFT for waveform reconstruction. Specifically, during training, we use the true phase to recover ground motion data and construct a waveform loss to better capture waveform shapes. In the generation stage, phase information is estimated using phase retrieval methods. Furthermore, to generate the earthquakes in which we are interested, we integrate physical parameters (i.e., earthquake magnitudes and depths, geospatial coordinates of sensors and earthquakes) as conditional variables into the CGM-GM framework using Multilayer Perceptron (MLP) layers. These conditional parameters are fundamental for understanding earthquake applications since ground motions vary spatially due to structural heterogeneity and the magnitudes reflect the energy released by rupturing. Fig. 2c illustrates that we can obtain spatial variations of ground motions at a given scenario of earthquake locations and magnitudes. Another motivation for using geospatial coordinates is to enable neural networks to implicitly learn the physical interactions, such as path, source, and site effects. The rupture distance $R_{rup}$ and incident angles $A_{epi}$ can be computed based on the coordinate information (i.e., latitude, longitude, and depth) of earthquake hypocenters and stations of interest. In this study, we have not incorporated variations in focal mechanisms due to our focus on the Hayward fault, where the majority of seismic events exhibit similar focal mechanisms. A more detailed discussion of this aspect is presented in the "Discussion" section.

### Waveforms and spatial continuity

Based on earthquake magnitudes and geospatial coordinates of earthquake sources and stations, the generator part of our CGM-GM framework produces physically consistent ground motion data. Figure 1e shows three representative comparisons between the generated ground motion waveforms and the true recordings. The selected cases involve seismic waveforms of similar magnitudes but with varying rupture distances, epicenter-station azimuths, and earthquake depths. Furthermore, we produce two random generations (blue) with the defined conditional variables to compare them with the corresponding observed data (red) in the time domain. The generated ground motion sequences effectively capture waveform shapes, frequency contents, peak values (e.g., waveform amplitude), and the arrival time, even for events with different rupture distances, depths, and magnitudes. For instance, in the first earthquake ($M = 2.51$), the model successfully captures the moderate amplitude P-wave packet around 14 s, followed by the large amplitude S-wave and surface wave packets starting at 22 s. Noticeably, the peak amplitudes of wavefields are well generated. More detailed investigations of these peak amplitudes and their spectra are presented in subsequent sections. Another advantage of generative modeling for ground motion waveforms is to perform uncertainty analysis, as discussed in Supplementary Section 2.3. For various earthquake scenarios, the mean curves of generated waveforms can capture dynamic characteristics of ground motion data, and the uncertainty regions show a good coverage of the ground truth. This indicates that our CGM-GM framework is effective and robust for generating earthquake ground motions.

A key aspect of our proposed generative model lies in its ability to approximate ground motions for arbitrary (future) earthquakes and sensor locations. To demonstrate it, we compute FAS maps across a specific region in the SFBA. FAS provides valuable insights into the spatial variability of frequency-dependent ground motion and informs seismic hazard assessments, structural design, and risk mitigation strategies. Specifically, using our CGM-GM framework, we generate the ground motions within a selected sub-region of the SFBA, which spans longitudinally from −122.50° to −121.35° and latitudinally from 37.27° to 38.07°. A uniform $100 \times 100$ spatial grid is sampled on a geographic map, yielding 10,000 station coordinates. We use the source parameters of an existing earthquake event that occurred at 3:16 a.m. on October 21, 2011. Our model is not intended to reproduce ground motions of this specific event; rather, the source parameter information is used as a representative case of an earthquake along the Hayward fault. Utilizing these conditions, 10,000 ground motion instances are generated and the FAS values are computed at each spatial location. The FAS map produced by our generative model at 10 Hz is shown in Fig. 2c. It corresponds to the median across 30 realizations, providing a stable and representative depiction of the model output. The generated median FAS map presents spatial continuity even between station pairs separated by large distances (e.g., in the southeast of the map). Moreover, we observe the FAS decay with respect to distances and its variation with azimuths, which validates the effectiveness of capturing spatial heterogeneity in ground motions by embedding geospatial coordinates of sources and stations into the CGM-GM framework. These characteristics are consistently seen across frequencies ranging from 2 to 15 Hz and under various earthquake scenarios, as detailed in Supplementary Section 2.6.

We conduct a comparative study between the generated results and baseline models specifically tailored for the SFBA[47–49], to assess the performance of our CGM-GM. The selected baseline models integrate methodologies from both ML and seismology. The first method is the empirical ground motion model (GMM), which is built from the observations and widely employed to predict ground motion intensities for seismic hazard analysis[47]. We use a state-of-the-art non-ergodic GMM[49] that incorporates location-specific effects to accurately represent the ground motion intensities. It is specifically built for the SFBA from the same dataset used in this study. The second one, termed CGM-baseline, uses the same CGM-GM architecture but only includes three conditional variables: earthquake magnitudes, source depths, and epicentral distances. Figure 2a, b illustrates the FAS results of the non-ergodic empirical GMM and CGM-baseline. The median FAS

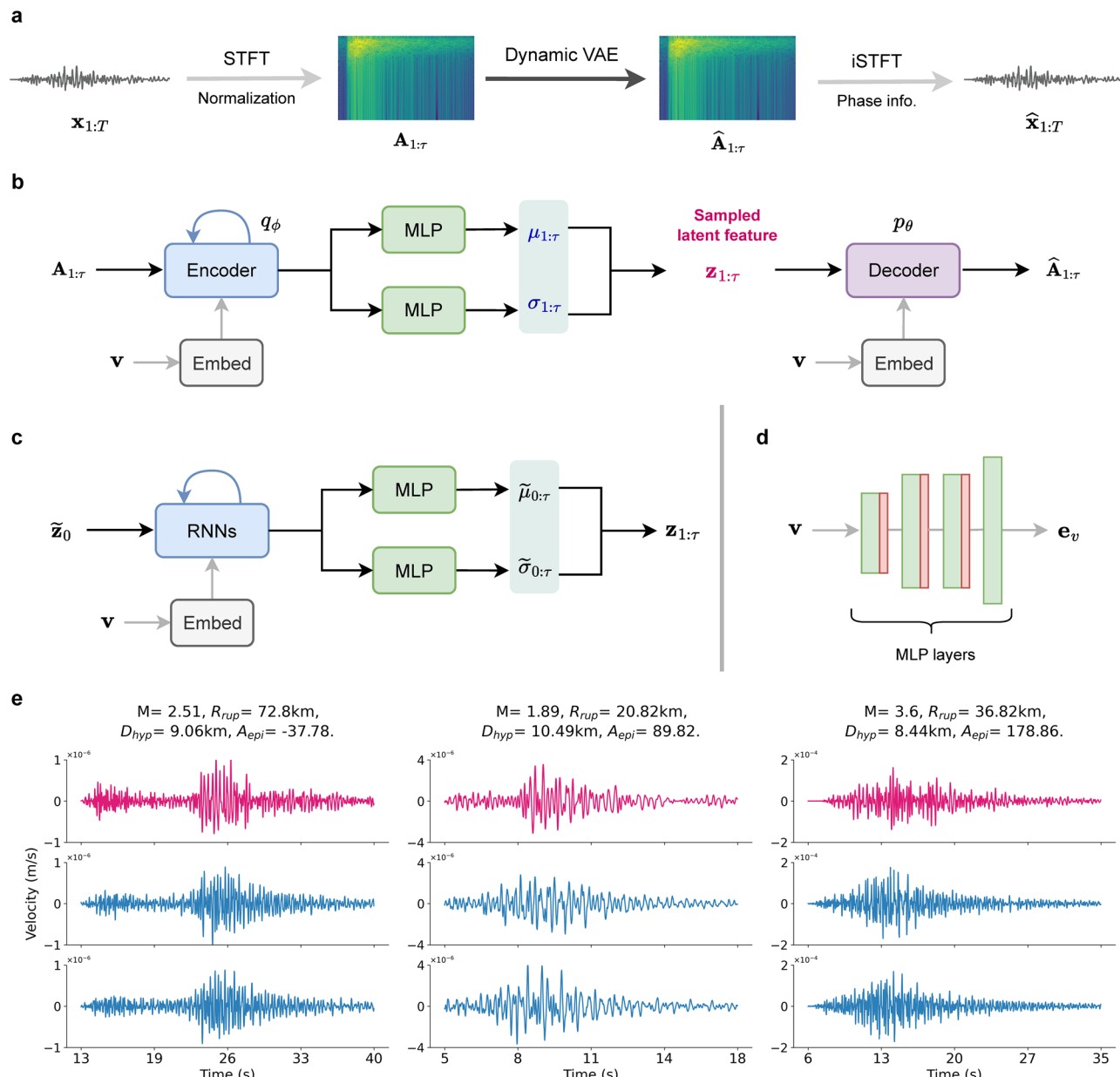

**Fig. 1 | Overview of our proposed CGM-GM for ground motion synthesis.**
**a** Illustrates the entire pipeline of the CGM-GM framework, where Short-Time Fourier Transform (STFT) is applied to extract time-frequency features and the dynamic Variational Autoencoder (VAE) model is designed for learning amplitude information. We leverage the true phase information for waveform reconstruction during training and consider phase retrieval methods in the stage of generation. Inverse STFT (iSTFT) is used for waveform reconstruction. Note that the ground motion sequence $\mathbf{x}(t)$ is in the time domain (with $T$ time step) and the amplitude spectrogram $\mathbf{A}(t, \omega)$ is in the time-frequency domain (with a time resolution of $\tau$). **b** Presents the network architecture of the dynamic VAE. $\mathbf{v}$ denotes the concatenation of multiple conditional variables, which are embedded into the VAE model. We use Multilayer Perceptron (MLP) layers to obtain the posterior mean $\boldsymbol{\mu}_{1:\tau}$ and variance $\boldsymbol{\sigma}_{1:\tau}$. **c** Shows the details of designing a sequential prior distribution,

where recurrent neural networks (RNNs) are used to incorporate dynamics into the model prior. **d** Displays the embedding module of conditional variables, where MLP layers (green blocks) and Rectified Linear Unit (ReLU) activation functions (red blocks) are used. $\mathbf{e}_v$ denotes the latent feature of conditional variables. **e** Shows the illustrative waveform comparison between the generations (blue) and the corresponding ground truth (red). It shows ground motion sequences of the H1 component with different pairs of earthquake magnitudes $M$, rupture distances $R_{rup}$, earthquake depths $D_{hyp}$, and epicenter-station azimuths $A_{epi}$. For each scenario, two waveforms are randomly generated given the same conditional variables. More examples of generated waveforms can be found in Supplementary Sections 2.1 and 2.2, including those showing moderate performance and the generations for the H2 component. Note that the waveforms are displayed based on their individual motion durations.

map generated by the CGM-baseline reveals a radial pattern, attributed to the constrained conditional embedding and the implicit assumption of a radially homogeneous Earth subsurface across the spatial domain. Hence, the contours of this FAS map are circular, showing geometrical spreading and average attenuation effects. In contrast, the CGM-GM and non-ergodic GMM capture more local features by incorporating

location-specific factors, which provide a more nuanced representation of the spatial variability in ground motions. For instance, both models predict larger motions in the southern region near San Jose compared to the CGM-baseline predictions. This is reasonable considering that the soft bay mud in the area amplifies the ground motions. However, certain discrepancies are observed in the spatial

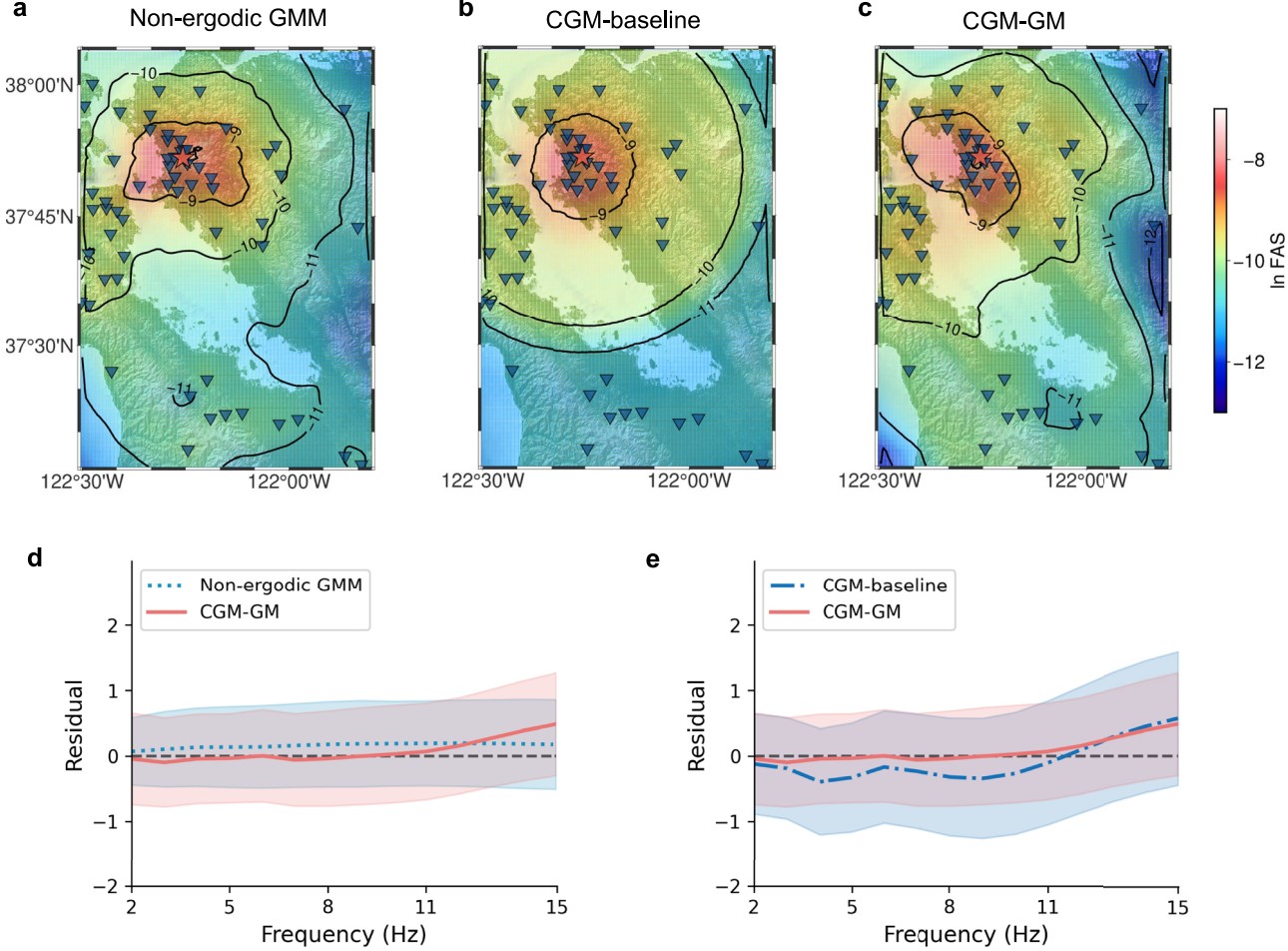

**Fig. 2 | Illustrative examples of generated Fourier Amplitude Spectra (FAS) maps. a–c** The FAS maps of non-ergodic ground motion model (GMM), CGM-baseline, and our CGM-GM at a frequency of 10 Hz. The red star and blue triangle denote the earthquake source and observation station, respectively. The seismic event is characterized by a magnitude of 3.84 and a depth of 7.94 km. The epicenter, denoted by a red star, is located at a geographic position with a latitude of $37°51.6'N$ and a longitude of $122°15.6'W$. A specific spatial region in the San Francisco Bay Area is selected for evaluation. Note that the generated FAS maps

from CGM-GM and CGM-baseline represent the median across 30 independent realizations. We provide more generations of FAS maps under various earthquake scenarios in Supplementary Section 2.6. **d, e** The FAS difference (Residual) between ground truth and the simulated samples from our generative model and the baseline models (non-ergodic GMM and CGM-baseline) for all earthquake recordings across the entire range of frequencies between 2 and 15 Hz. The discrepancy is calculated by the logarithmic residual of FAS values. The solid line and the shaded area denote the mean curves and the region of mean ± std.

distributions of FAS maps derived from our generative model and the non-ergodic GMM. The CGM-GM predicts slightly larger motions in the northwest region near San Francisco than the non-ergodic GMM, which is also plausible due to the presence of Bay Mud in the area. The CGM-GM predictions exhibit significant variations across the San Andreas and Hayward faults. The sharp lithological changes lead to substantial differences in ground motions on either side of the faults[50,51]. The first-order agreement between the CGM-GM and non-ergodic GMM demonstrates the validity of our framework in learning hidden ground motions. In Supplementary Section 2.11, we further provide a detailed analysis of the spatial correlation in the FAS values generated by the CGM-GM model.

To further evaluate the accuracy of all models, we investigate their performance against true ground motion recordings for all earthquake events. Figure 2d, e illustrates the averaged FAS differences (Residuals) in the form of a natural logarithm between the true recordings and simulated samples from the non-ergodic GMM, the CGM-baseline, and the CGM-GM model across the entire frequency band of [2,15] Hz. This comparison includes the mean values and their associated uncertainties, represented by one standard deviation (std). Note that our model does not explicitly differentiate between epistemic uncertainty (due to

data scarcity) and aleatory variability (arising from randomness of earthquakes)[52]. Overall, our generative model and the non-ergodic GMM show similar residuals. At frequencies between 3 and 11 Hz, the CGM-GM slightly outperforms the non-ergodic GMM, though its performance declines at frequencies above 11 Hz. The uncertainty ranges for both models overlap significantly. These findings indicate that the CGM-GM performs comparably to the state-of-the-art non-ergodic GMM. In comparison to the CGM baseline, our CGM-GM demonstrates reduced misfits and uncertainty over the entire frequency, confirming the importance of incorporating spatial heterogeneities into generative models.

## Amplitude spectra

We further assess the performance of our CGM-GM in the frequency domain, specifically analyzing FAS values and their frequency distributions versus rupture distances. To present a comprehensive evaluation, we select diverse ranges of earthquake magnitudes and geospatial coordinates, generating 100 random samples for each set of conditional variables. For instance, as shown in the top-left part of Fig. 3a, we select the range of earthquake magnitudes in [1.75, 2.25], the rupture distances in [10, 20] km, and the earthquake depths in

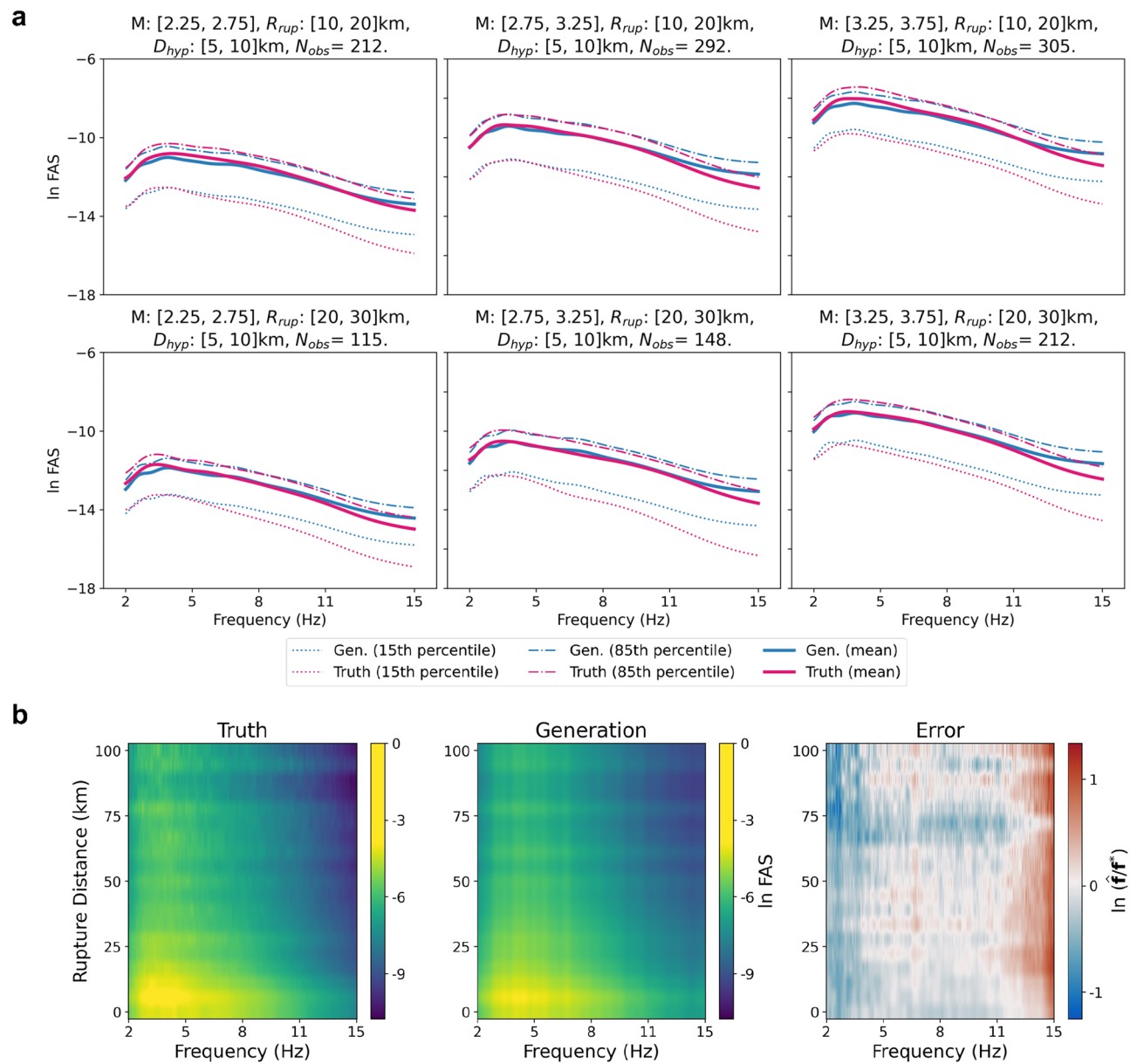

**Fig. 3 | The comparative analysis of amplitude spectra information between the ground truth and the generated samples. a** The comparison of Fourier Amplitude Spectra (FAS) results from true seismic recordings and the generated waveforms across diverse conditional variables. $N_{obs}$ denotes the number of observations. The earthquake depths are within a fixed range in all plots. We show FAS results at the 15th percentile, mean, and 85th percentile. "Gen." denotes the results from generations. **b** Compares amplitude spectra heatmaps from ground truth and generated data. The error heatmap is based on the logarithmic division of the ground truth $\mathbf{f}^*$ and the generations $\widehat{\mathbf{f}}$.

[5, 10] km, where 63 ground motion sequences in our field dataset. We obtain the statistical values (e.g., 15th percentile, mean, and 85th percentile) for ground truth and generation results using 63 and 6300 samples, respectively.

Figure 3a shows the variations of FAS values across diverse earthquake magnitudes and rupture distances. Generally, the FAS results simulated by our CGM-GM model align well with the true FAS curves, especially for mean curves. However, we also observe that the CGM-GM model tends to slightly under-predict at the low-frequency part ([2,3] Hz) and over-estimate at the high-frequency region ([13,15] Hz). This is due to the inductive bias of VAE models, where the learned amplitude information is over-smoothing in the logarithmic space. Similarly, this phenomenon is also seen in Fig. 3b, which illustrates a heatmap of frequency distributions versus rupture distances. The error is computed with the division of the natural logarithm of

generated FAS $\widehat{\mathbf{f}}$ and ground truth $\mathbf{f}^*$, which is given by $\ln(\widehat{\mathbf{f}}/\mathbf{f}^*)$. The heatmap of the generation results exhibits a generally good alignment with that of the ground truth data. The discrepancies primarily manifest in low-frequency and high-frequency parts. However, the magnitude of logarithmic division errors remains within an acceptable range (approximately [−1.1, 1.2]). Additionally, we provide the comparisons between the generated and true FAS values across different earthquake depths in Supplementary Section 2.4, and discuss the model performance for the H2 component in Supplementary Section 2.5.

## P and S arrival times and durations
To further evaluate the quality of synthesized waveforms, we analyze the statistical properties of amplitudes and arrival times of the generated ground motions in the time domain. Statistical analysis of ground motion data is essential for developing predictive models. In

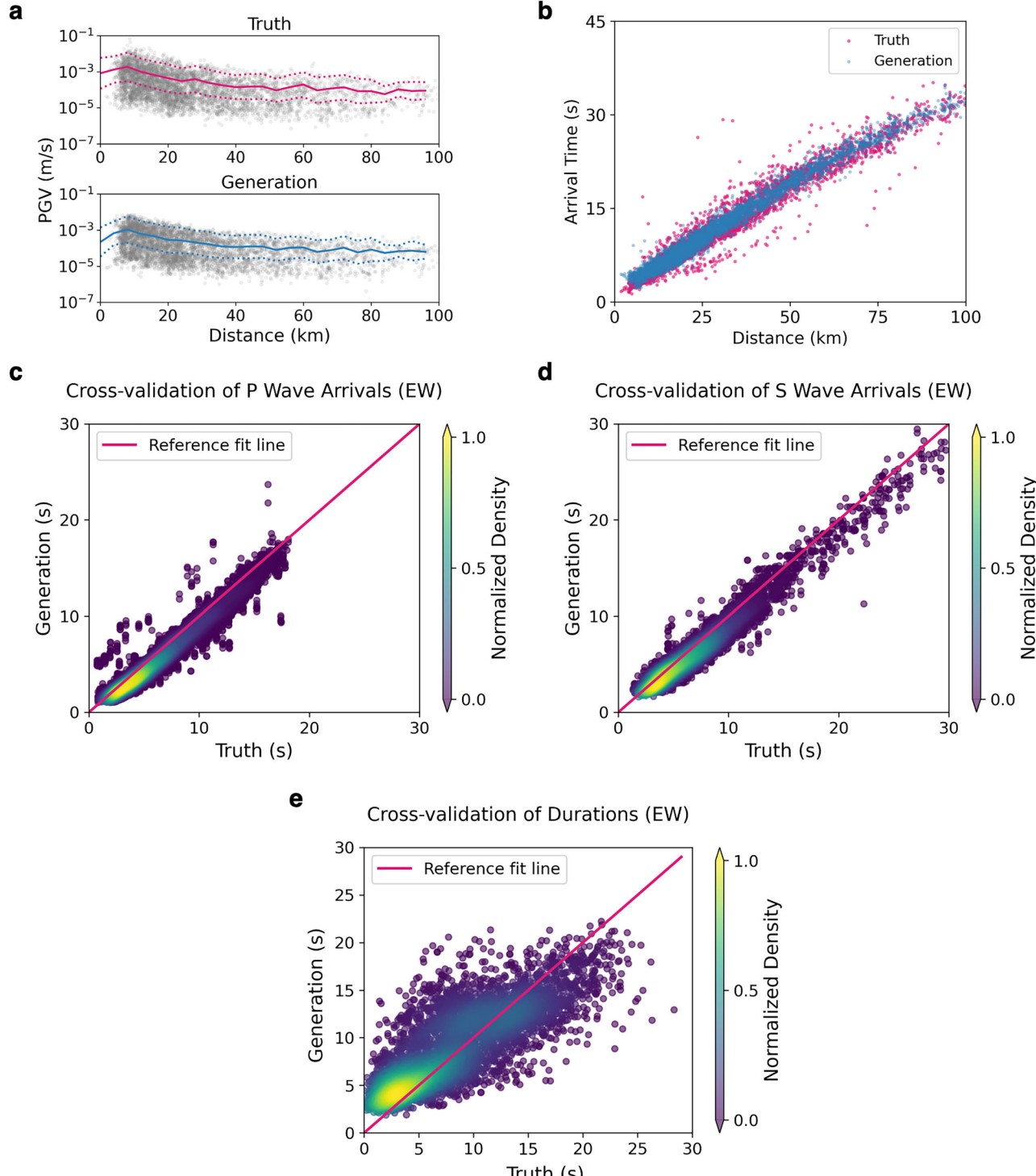

**Fig. 4 | The statistical evaluation of the generated waveforms. a** Presents the peak ground velocity (PGV) distribution versus rupture distances. The solid lines denote the mean curves and the dashed lines show the boundaries of mean ± std. **b** Is the scatter plot of the arrival time versus distances for generated and truth waveforms. **c**, **d** Show the cross-validations of the arrival time of P and S waves in the East-West (EW) direction, respectively, including scatter plots colored by density and reference fit lines. **e** represents the cross-validation of the shaking durations in the EW direction. The density is normalized between 0 and 1.

this study, we focus on investigating the relationship between rupture distances and three evaluation metrics: the PGV distribution, the peak arrival time (i.e., arrival times of the wavelet of the PGV), and the arrival times of P- and S- waves. Specifically, we implement 100 realizations of ground motion generation with consistent conditional variables from the SFBA dataset. All the statistical analyses are based on these

multiple-run generations. Firstly, the PGV distribution across the rupture distances is visualized in Fig. 4a. The solid lines and dashed lines denote the mean curves and the boundaries of mean ± standard deviation (std), and the dark points present the corresponding PGV values of data samples. We observe that the generated samples effectively capture the distribution of PGV values, including

magnitudes and associated uncertainty regions, across diverse rupture distances. It implies that the generations can match the ground truth data well. Moreover, we analyze the arrival of seismic waves. As shown in Fig. 4b, the generated results present a high correlation with the ground truth data in terms of the arrival time versus rupture distances. To gain a deeper insight into the performance of our generative model, we further investigate the performance of capturing the arrival time of P and S waves in the EW direction. Specifically, we use the PhaseNet[53] to pick the arrival time of two types of waves with a probability larger than 50%. Figure 4c, d show the cross-validations between the generated and true arrivals, including 3138 and 909 samples for P and S waves, respectively. The result demonstrates a good agreement and validates the effectiveness of our proposed generative modeling method. Additionally, we investigate the durations of the simulated ground motion waveforms. We compute a significant duration $D_5 - D_{95}$ based on area intensities[54], which is another important ground motion intensity metric in engineering and seismic hazard analysis. Figure 4e presents the cross-validations of the durations in the EW direction, which exhibits greater scatter than arrival time results illustrated in Fig. 4c, d. It indicates the difficulties of accurately generating and measuring seismic durations. Moreover, we observe a bias primarily for durations within the [0, 3] second range, corresponding to smaller-magnitude earthquakes. Due to lower S/N ratios in such cases, it is challenging to precisely measure the durations from actual seismic data since the relatively high noise level can artificially reduce the measured duration compared to the actual signal duration. Note that the generative model does not explicitly simulate noise, resulting in inherently higher S/N ratios. This potentially amplifies the discrepancy. Nonetheless, despite these limitations, the overall duration trends between observed and generated data demonstrate a good general correlation.

### Generalization to larger-magnitude scenarios

Large-magnitude earthquakes produce intense ground shaking across wide areas, posing significant hazards to infrastructure and communities. In this part, we show the magnitude scaling performance of our CGM-GM model compared to empirical GMMs, including a non-ergodic GMM for small-magnitude earthquakes in Section Empirical GMMs and the LA21 model[55] for large-magnitude earthquakes in California. Since our model is only trained on small-magnitude earthquakes ($M < 4$), directly generalizing to large-magnitude scenarios will be challenging. Therefore, we fine-tune our pre-trained model on two larger-magnitude earthquake data: (i) the 2014 Napa earthquake with a magnitude of 6.02 and depth of 11.0 km, and (ii) the 2018 Berkeley earthquake with a magnitude of 4.38 and depth of 12.3 km. In Fig. 5a, we present the scaling performance up to a magnitude of 8, occurring at a hypocenter depth of 7.94 km, with geographic coordinates of latitude $37°51.6'N$ and longitude $122°15.6'W$ (the same location as shown in Fig. 2). The magnitude is varied from 1 to 8, and waveforms are generated at 73 real-world stations.

The CGM-GM predictions exhibit a reasonable increase in FAS with increasing magnitude, but deviations from the LA21 model remain challenging. First, the CGM-GM predictions remain lower than those predicted by the LA21 model for magnitudes between 4 and 7. This discrepancy may arise because our model is trained mostly on small-magnitude events, where the earthquake source rupture processes can be approximated as point sources. In contrast, large-magnitude events require consideration of finite rupture planes, which introduces deviations in the predictions. Furthermore, the CGM-GM does not account for nonlinear effects, which are typically weak in small-magnitude events but become significant at higher shaking intensities. As a result, the model produces a near-linear increase in FAS with magnitude, unlike the LA21 model, which captures the saturation behavior observed in large-magnitude events. As shown in Supplementary Section 2.10, the fine-tuning strategy improves the model

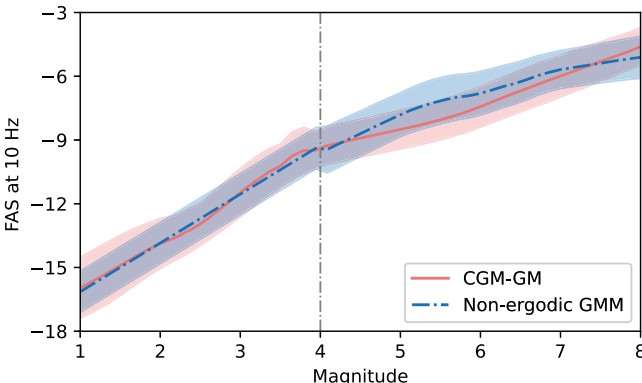

**Fig. 5 | Comparison of the magnitude scaling performance of our CGM-GM model against empirical ground motion models (GMMs) using Fourier Amplitude Spectra (FAS) (10 Hz).** For small-magnitude earthquakes ($M < 4$), we benchmark the CGM-GM model against the non-ergodic GMM developed for the San Francisco Bay Area[49]. For large-magnitude earthquakes ($M > 4$), we compare it with the non-ergodic LA21 model[55]. The solid line and the shaded area denote the mean curves and the region of mean ± std.

performance by enhancing FAS values, which further supports this observation. To enhance model performance, we plan to explore fine-tuning using larger datasets and incorporate out-of-distribution generalization methods[56]. More details can be found in Supplementary Section 2.10.

## DISCUSSION

In this section, we discuss the inherent sparsity in real-world earthquake datasets and the selection of conditional variables due to their significant effects on the performance of generative modeling. Due to practical constraints, the spatial distribution of seismic stations is often sparse and non-uniform across the geospatial domain. This sparsity leads to uneven data coverage and gaps in the observation data, which poses challenges for generative models in accurately capturing the underlying spatial heterogeneity. This task is intrinsically complicated since it requires the GMMs to infer spatial variability and site-specific effects without direct observation data. Therefore, the generative models might exhibit artifacts in producing FAS maps for areas lacking measurement data.

In our generative model, focal mechanisms are not included as conditional variables for three reasons. Firstly, our earthquake data are mostly concentrated around the Hayward and San Andreas Faults, where these earthquakes predominantly exhibit similar focal mechanisms (i.e., the right lateral faulting). Importantly, our primary interest is also in generating waveforms along the Hayward Fault with consistent focal mechanisms, which indicates no domain shift for the earthquake rupturing processes. The second reason is the data limitation. As mentioned, there is insufficient variation in the focal mechanisms of the recorded earthquakes to effectively train the network for arbitrary focal mechanism parameters (e.g., fault strike). Typically, generative models work well when the generated data interpolates within the range of the training data, as we have demonstrated for source and receiver locations. Incorporating focal mechanisms as a variable would only yield reliable generations within very limited focal mechanisms. Lastly, the use of geographical coordinates as conditional variables allows us to handle various focal mechanisms, while implicitly assuming that focal mechanisms remain consistent (although magnitudes may vary) for earthquakes occurring at the same geospatial location. In one of our follow-up studies, we use seismic records from the Geysers geothermal field[57,58]. Earthquakes in this region exhibit a range of mechanisms, including normal, reverse,

and strike-slip faulting[59]. With appropriate parameter settings, the CGM-GM model is capable of capturing focal mechanism variations.

Our model learns to capture the properties, such as spatially continuous FAS, through a combined representation of source, site, and path effects. Nevertheless, these effects are not explicitly separated in the current conditional generative framework. Developing methods for their explicit characterization will be an important direction for future research. Another potential limitation lies in the lack of constraints of path effects. The current generative modeling framework only incorporates spatial information as conditional variables without explicitly considering the correlation between different pairs of stations and sources. The implicit learning of paths can be achievable since the training waveform data naturally contains the path effect. When numerous waveforms are available for training, the generative model should be able to learn the path effects. For instance, Gatti et al.[60] has demonstrated the benefits of integrating physics-based simulation data with real-world observations to improve model performance in earthquake ground motion prediction, especially learning the path effect. However, in reality, the data is often limited and we may need an explicit way to include the path effects. To address this issue, we would like to investigate the incorporation of the spatial correlation, such as Matérn covariance function[61,62], into generative modeling. Another consideration is the phase spectra, which play a critical role in capturing the complexity of waveform time series. Nevertheless, maintaining phase continuity poses a greater challenge than ensuring spatial continuity of amplitudes due to the rapid variability of phase characteristics. In addition, using phase retrieval methods for waveform reconstruction may lead to inaccurate phase information in the generated samples due to the intrinsic difficulty in such ill-posed inverse problems. VAE models are known to generate relatively smooth amplitude information, which might introduce numerical artifacts during phase retrieval. This may partially explain the occurrence of high-frequency spikes, as shown in Supplementary Section 2.1. Hence, we will consider building a generative model directly in the time domain instead of the time-frequency domain to avoid using phase retrieval methods. Another direction is exploring specific techniques, such as generalized variance parameterization[63] and heavy-tailed distributions[64], to mitigate the over-smoothing issue in VAE models.

Our proposed generative modeling framework is user-friendly and stable to train. It presents effectiveness and efficiency in producing synthetic ground motions for various earthquake scenarios with different magnitudes and geographical regions in the SFBA. The central part is that our CGM-GM can capture the spatial heterogeneity of FAS without explicit physics constraints. We conduct a comparative assessment of our CGM-GM against baseline models, including the CGM-baseline and state-of-the-art non-ergodic GMMs. The results demonstrate that our method performs comparably to the advanced non-ergodic GMMs. Moreover, we comprehensively evaluate the performance of the model on generative modeling for ground motion synthesis in both time and frequency domains. For the assessment in the time domain, the ground motion samples generated by CGM-GM show excellent capability of capturing waveform shapes, PGVs, and arrival times. For the evaluation in the frequency domain, the FAS values exhibit good agreement between the observed and generated data. Overall, we anticipate that the promising results of scientific generative AI modeling for ground motion synthesis will encourage researchers to explore this area, and the potential issues we have identified will enable the development of more effective methods for enhancing generation quality.

## Methods

In this section, we present the technical details of our generative model in the context of ground motion simulation. The entire framework is shown in Fig. 1a–d, including the forward and inverse STFT, the network design of dynamic VAE, the sequential model prior, and the embedding of conditional variables. Here, we use the dataset from the H1 component as an illustrative example, and the results on the H2 component are provided in Supplementary Sections 2.2 and 2.5.

### Forward and inverse STFT

STFT has been widely applied in audio processing[65,66] and seismic data analysis[67–69]. It defines a valuable category of time-frequency distributions[70] that describe the amplitude and phase relationships with respect to time and frequency for any signal. This is achieved by repeatedly applying the Fourier transform within specific time windows. Typically, we use sliding time windows with overlaps to capture signals throughout the entire time domain. Therefore, leveraging STFT in generative modeling[71,72] facilitates the extraction of time and frequency information $\mathbf{A}$ from the ground motion sample $\mathbf{x}$. Although the time-frequency resolution of STFT is fixed in the entire time and frequency domains with the chosen time window length, the inverse STFT is relatively stable compared to other 2D spectral decomposition methods, such as the continuous wavelet transform.

The inverse STFT is employed to reconstruct waveforms $\hat{\mathbf{x}}$ from amplitude spectrograms $\hat{\mathbf{A}}$. During the generation of artificial ground motions, phase retrieval methods are used to estimate the phase information. To be more concrete, these approaches estimate the missing phase information from available amplitude measurements and then recover the timing and shape of seismic waveforms in earthquake ground motion analysis. Various mathematical techniques are leveraged for phase retrievals, such as iterative algorithms and optimization frameworks. For instance, the Griffin-Lim algorithm[73] is widely used thanks to its simplicity and effectiveness. It iteratively refines estimates of seismic wave phases to minimize the discrepancy between the original and reconstructed signals. On the other hand, the Alternating Direction Method of Multipliers (ADMM)[74] is a versatile optimization technique that has gained traction in various scientific domains, including ground motion synthesis[36]. ADMM leverages a convex relaxation framework to decompose complex optimization problems into simpler subproblems and solve the augmented Lagrangian function iteratively. In this paper, we leverage the dynamic VAE model to learn amplitude information and employ the Griffin-Lim algorithm for phase retrieval due to its simplicity. This strategy is chosen over generating phase information with the VAE model, as phase signals are complex and amplitude evolution is relatively smooth over time.

### Dynamic VAEs

We consider a dynamic VAE architecture since it is specifically designed to model sequential data with temporal correlations by extending from standard VAEs[41,46,75]. Although VAE models have achieved great success in image processing tasks, the absence of explicit temporal modeling in standard VAEs hinders their effectiveness in tackling time series and audio data[76].

The dynamic VAE models focus on a sequence-to-sequence mode for encoding and decoding. Namely, the latent variable is constructed in the form of a temporal sequence instead of a static vector. Let us consider a time sequence $\mathbf{x}_{1:T} = \{\mathbf{x}_t \in \mathbb{R}^N\}_{t=1}^T$, where $T$ is the sequence length. Dynamic VAE models typically yield a sequence of latent variables $\mathbf{z}_{1:T} = \{\mathbf{z}_t \in \mathbb{R}^l\}_{t=1}^T$. Therefore, the joint distribution of latent and observed sequences can be reformulated as

$$p_{\boldsymbol{\theta}}(\mathbf{x}_{1:T}, \mathbf{z}_{1:T}) = \prod_{t=1}^{T} p_{\boldsymbol{\theta}}(\mathbf{x}_t|\mathbf{x}_{1:t-1}, \mathbf{z}_{1:t}) p_{\boldsymbol{\theta}}(\mathbf{z}_t|\mathbf{x}_{1:t-1}, \mathbf{z}_{1:t-1}). \quad (1)$$

Eq. (1) is a generalized version that describes the generative process in dynamic VAEs. Researchers usually resort to state-space models (SSMs) to simplify the dependencies in conditional distributions of Eq. (1)[76]. One of the most commonly used SSM families is RNNs, which

are specifically designed network architectures for handling sequential data and capturing temporal dependencies among data points. However, a significant challenge in training vanilla RNNs is the vanishing and exploding gradient problem. The reason behind this phenomenon is that the gradients can either shrink exponentially (vanishing gradients) or grow exponentially (exploding gradients) when RNNs propagate information through time. This instability hinders effective training and prevents the network from learning long-term dependencies. To alleviate such issues, gating-based RNNs, such as LSTM[77] and GRU[78], are proposed to control the flow of information through the network.

By incorporating the auto-regressive recurrence into dynamic VAE models, the generative process can be simplified as

$$p_{\boldsymbol{\theta}}(\mathbf{x}_{1:T}, \mathbf{z}_{1:T}) = \prod_{t=1}^{T} p_{\boldsymbol{\theta}}(\mathbf{x}_t|\mathbf{z}_t)p_{\boldsymbol{\theta}}(\mathbf{z}_t|\mathbf{z}_{1:t-1}). \qquad (2)$$

Moreover, the approximate posterior can be reformulated as

$$q_{\boldsymbol{\phi}}(\mathbf{z}_{1:T}|\mathbf{x}_{1:T}) = \prod_{t=1}^{T} q_{\boldsymbol{\phi}}(\mathbf{z}_t|\mathbf{z}_{1:t-1}, \mathbf{x}_{1:t}), \qquad (3)$$

where $q_{\boldsymbol{\phi}}(\mathbf{z}_{1:T}|\mathbf{x}_{1:T})$ works as an inference model for the latent sequence $\mathbf{z}_{1:T}$ from the observed sequential data.

The training of a dynamic VAE model involves optimizing the evidence lower bound (ELBO) on the marginal likelihood of the observed data $\mathbf{x}_{1:T}$. For a given data sample $\mathbf{x}_{1:T}$, the marginal likelihood is defined as

$$\begin{aligned} \log p_{\boldsymbol{\theta}}(\mathbf{x}_{1:T}) &= D_{KL}(q_{\boldsymbol{\phi}}(\mathbf{z}_{1:T}|\mathbf{x}_{1:T})||p_{\boldsymbol{\theta}}(\mathbf{z}_{1:T}))||p_{\boldsymbol{\theta}}(\mathbf{z})) + \mathcal{L}(\boldsymbol{\theta}, \boldsymbol{\phi}; \mathbf{x}_{1:T}) \\ &\geq \mathcal{L}(\boldsymbol{\theta}, \boldsymbol{\phi}; \mathbf{x}_{1:T}), \end{aligned} \qquad (4)$$

where $D_{KL}(\cdot)$ denotes the Kullback-Leibler (KL) divergence between the approximate and true posterior distributions. $D_{KL}(\cdot)$ is a non-negative term. Therefore, $\mathcal{L}(\boldsymbol{\theta}, \boldsymbol{\phi}; \mathbf{x}_{1:T})$ represents the ELBO, which is given by

$$\begin{aligned} \mathcal{L}(\boldsymbol{\theta}, \boldsymbol{\phi}; \mathbf{x}_{1:T}) &= \mathbb{E}_{q_{\boldsymbol{\phi}}(\mathbf{z}_{1:T}|\mathbf{x}_{1:T})}[\log p_{\boldsymbol{\theta}}(\mathbf{x}_{1:T}|\mathbf{z}_{1:T})] \\ &\quad - D_{KL}(q_{\boldsymbol{\phi}}(\mathbf{z}_{1:T}|\mathbf{x}_{1:T})||p_{\boldsymbol{\theta}}(\mathbf{z}_{1:T})). \end{aligned} \qquad (5)$$

The first and second terms on the right-hand side (RHS) are reconstruction loss and a KL-divergence term, respectively. The KL-divergence works as a regularizer for $\boldsymbol{\phi}$ that promotes the approximate posterior $q_{\boldsymbol{\phi}}(\mathbf{z}_{1:T}|\mathbf{x}_{1:T})$ to closely resemble the prior $p_{\boldsymbol{\theta}}(\mathbf{z}_{1:T})$.

### Network design

As shown in Fig. 1b, our dynamic VAE model is trained on amplitude information $\mathbf{A}_{1:\tau}$ ($\tau < T$) to facilitate the learning of time-frequency features of ground motion data. In the CGM-GM framework, we incorporate a GRU layer[78] into the Encoder to learn the dynamics. Subsequently, two MLP layers yield the posterior mean $\boldsymbol{\mu}_t$ and variance $\boldsymbol{\sigma}_t$ at each time stamp $t$. Hence, the reparameterization trick to produce the posterior sequence $\mathbf{z}_{1:\tau}$ can be written as

$$\mathbf{z}_t \sim \mathcal{N}(\boldsymbol{\mu}_{\boldsymbol{\phi}}(\mathbf{A}_t), \boldsymbol{\sigma}_{\boldsymbol{\phi}}^2(\mathbf{A}_t)). \qquad (6)$$

The sampled latent sequence is fed into the Decoder part to obtain the reconstructed amplitude information $\widehat{\mathbf{A}}_{1:\tau}$. In the training stage, we directly leverage the true phase information to reconstruct the time series $\widehat{\mathbf{x}}_{1:T}$. In the generation stage, we conduct the inverse STFT procedure to generate the artificial ground motion sequence $\widehat{\mathbf{x}}_{1:T}$ by utilizing the phase retrieval method, i.e., the Griffin-Lim algorithm[73]. Furthermore, we design a sequence of Gaussian distributions to serve as a dynamic model prior[79], as shown in Fig. 1c. The mathematical

formulation is given by

$$p_{\boldsymbol{\theta}^*}(\mathbf{z}_t|\mathbf{z}_{1:t-1}) = \mathcal{N}(\boldsymbol{\mu}(\mathbf{z}_{1:t-1}; \boldsymbol{\theta}^*), \boldsymbol{\sigma}^2(\mathbf{z}_{1:t-1}; \boldsymbol{\theta}^*)). \qquad (7)$$

The model prior distribution (i.e., mean and variance) is learned through a sub-network architecture $\boldsymbol{\theta}^*$, where another GRU layer and two MLP layers are employed for capturing the underlying dynamics. Moreover, to better capture the waveform shapes, we incorporate another waveform loss to the total loss function apart from the reconstruction loss of amplitude information and the KL-divergence in Eq. (5). The waveform loss is constructed by calculating the difference between the true waveforms and the recovered counterparts using the true phase information in the training. The final loss function is defined as

$$\begin{aligned} \mathcal{L}(\boldsymbol{\theta}, \boldsymbol{\phi}; \mathbf{x}_{1:T}) &= \mathbb{E}_{q_{\boldsymbol{\phi}}(\mathbf{z}_{1:\tau}|\mathbf{A}_{1:\tau})}[\log p_{\boldsymbol{\theta}}(\mathbf{A}_{1:\tau}|\mathbf{z}_{1:\tau})] \\ &\quad - \beta \cdot D_{KL}(q_{\boldsymbol{\phi}}(\mathbf{z}_{1:\tau}|\mathbf{A}_{1:\tau})||p_{\boldsymbol{\theta}}(\mathbf{z}_{1:\tau})) \\ &\quad + \alpha \cdot \mathbb{E}_{q_{\boldsymbol{\phi}}(\mathbf{z}_{1:\tau}|\mathbf{x}_{1:\tau})}[\log p_{\boldsymbol{\theta}}(\mathbf{x}_{1:T}|\mathbf{z}_{1:\tau})]. \end{aligned} \qquad (8)$$

Here, $\alpha$ and $\beta$ are two weighting coefficients for waveform loss in the time domain and the KL-divergence term. The incorporation of waveform loss facilitates the capturing of peak values and the generation of realistic earthquake ground motion shapes.

Generally, we clarify the training (reconstruction) and inference (generation) processes of our CGM-GM as follows. In the training stage, the dynamic VAE model is optimized to reconstruct input waveforms from their encoded latent representations $\mathbf{z}_{1:\tau}$. This process enables the latent space to learn a meaningful representation of the underlying seismic data. The prior distribution acts as a regularization term through the KL-divergence, enforcing the latent space distribution to remain close to the prior. The reconstruction loss and the KL divergence regularization guide the model in learning a smooth and structured latent space and facilitate effective sampling during the generative process. Unlike the training stage for reconstruction, the inference process (generation) does not rely on existing input waveforms to synthesize ground motion data. During inference, ground motion waveforms are generated by directly sampling from the prior distribution and passing it through the decoder to produce new waveforms. By incorporating conditional variables (e.g., geospatial coordinates, earthquake magnitudes, and depths) into the model prior and the decoder, the entire framework ensures the generation of consistent and realistic ground motion waveforms for arbitrary earthquake scenarios.

### Embedding conditional variables

We design an embedding module, consisting of a stack of MLP layers, to integrate the conditional variables into the framework. The illustration of embedding conditional variables is presented in Fig. 1d. To fully encode physical knowledge into the generative process, we apply this shareable embedding module to the encoder, the decoder, and the model prior. For the embedded physical variables, we use earthquake magnitudes and the geospatial coordinates of earthquake sources and stations. The primary rationale for incorporating geospatial coordinates is that many physical properties, such as source and site effects, are inherently coordinate-based and crucial for ground motion simulation. Neural networks can implicitly learn representations of the underlying physics and kinematics with the coordinates as inputs[21,80]. Note that, in previous papers[35,36], the researchers also use the information $Vs_{30}$. Given that $Vs_{30}$ is an empirical parameter based on geological coordinates, the embedding network should be capable of implicitly capturing these velocity properties with coordinate information as inputs. $Vs_{30}$ is a proxy correlated with site-specific

**Table 1 | Conditional variables used in CGM models**

| Model | Conditional variables |
|---|---|
| CGM-baseline | Earthquake magnitudes, source depths, and epicentral distances |
| CGM-GM | Earthquake magnitudes, geospatial coordinates of earthquake sources and stations |

ground-motion properties, and it is often used in the development of ground-motion models. However, estimations of $Vs_{30}$ are not always available at all sites and can present significant uncertainties. Thus, we choose not to integrate $Vs_{30}$ into our model as a parameter and instead estimate local site conditions directly from ground motion data with conditional generative modeling.

For comparison, the CGM-baseline employs earthquake magnitudes, source depths, and epicentral distances as conditional variables. The details of conditional variables for CGM-GM and CGM-baseline are listed in Table 1.

### Data selection
A total of 626,423 recordings are collected that spans from 10 s prior to the event time to 60 s after the event time, leading to each recording of 70 s with 7000 time steps per component. We perform a selection procedure to ensure only recordings with an acceptable S/N ratio are used. Specifically, the first 10 s of each recording are considered as noise, while the subsequent 60 s are analyzed as earthquake signals. The Fourier transforms of both the noise and the signal are calculated up to the Nyquist frequency of 50 Hz, and the Fourier amplitudes are smoothed using the Konno-Ohmachi window procedure[81] with a smoothing parameter $bexp = 20$[81]. The S/N ratio is computed for each recording by comparing the amplitude spectra of the signal and noise. We keep those recordings with an S/N ratio exceeding 3 across the frequency range of 2–15 Hz. Hence, approximately 15,000 recordings per component are retained. Additionally, we conduct a rigorous visual inspection for each time series to exclude those with equivalent noise levels in the first 10 seconds and the last 60 s. This process results in 5, 108 and 5, 301 recordings available for horizontal components H1 and H2, respectively. After selecting the final recordings, a bandpass filter is applied over the frequency range of interest in [2,15] Hz.

Figure 6a, b presents the distribution of earthquake depth and the scatter plot of the corresponding magnitudes and rupture distances, respectively. The locations of selected stations and events for the H1 component are shown in Fig. 6c, d. Note that our methods are applicable to diverse earthquake ground motion datasets (i.e., different magnitude ranges) and geographical regions. More complete information about the dataset can be found in ref. 82.

### Implementations
This part includes the data preprocessing and the training implementations. The last 60 seconds of the SFBA dataset are selected for ground motion generation. The dataset is then split into {80%, 20%} for training and testing, respectively. Namely, the number of training samples is 4086 for the experiments. Firstly, we use STFT to obtain the time-frequency information from the ground motion data since the CGM-GM model focuses on learning the amplitude spectrograms. The window length and hop length are set as 160 and 46, respectively. Therefore, the amplitude spectrogram has a size of $81 \times 131$, where 81 is the frequency range and 131 denotes the sequence length. The window length of 160 is equivalent to 1.6 s. We use signals between [2,15] Hz for our analysis, and this window length contains three wavelets for the lowest frequency to resolve signals at this frequency. The hop length is more arbitrary, and Welch[83] recommends using a length shorter than half of the window length. Due to the computational cost, we use 46 in this study. We add a minimum threshold of $10^{-10}$ for amplitude spectrograms and then convert them into the logarithmic space. Furthermore, the logarithmic time-frequency coefficients are normalized

between 0 and 1. For conditional variables, we consider the earthquake magnitudes and geospatial coordinates of sources and stations. Each variable is scaled within [0, 1] separately.

For the model parameters in CGM-GM, we consider a 3-layer MLP with feature sizes of [32, 32, 16] for embedding conditional variables. The label embedding module, as shown in Fig. 1d, is kept fixed for different parts of the network. For the Encoder component, we use one GRU layer with a hidden dimension of 144 and two independent 2-layer MLPs with dimensions of [64, 32] to obtain the sequential latent variables. Note that the outputs from the encoder part, $\mu_{1:\tau}$ and $\sigma_{1:\tau}$, have the same sequence length $\tau$ as the input amplitude spectrogram $\mathbf{A}_{1:\tau}$. For the decoder, a stack of MLP layers with feature sizes of [128, 128, 81] is employed for reconstructing the amplitude information. For the sequential prior model, we use one GRU layer with a hidden dimension of 32 and two independent linear layers to get the sequential prior variables. This model architecture contains 0.17 million parameters.

We use the Adam[84] optimizer to train the proposed method for 5000 epochs. The learning rate is set as $8 \times 10^{-4}$ initially and decays every 100 steps with a ratio of 0.99. The weight decay parameter is set as $5 \times 10^{-5}$ and the batch size is 128. Furthermore, we leverage the grid search to select the hyperparameters in the loss function. The optimal parameters are defined as $\alpha = 0.5$ and $\beta = 0.003$.

### Empirical GMMs
Empirical ergodic and non-ergodic GMMs for velocity FAS values are developed to evaluate the ground motions generated by our CGM-GM. Specifically, ergodic GMMs focus on the average scaling of ground motions, and non-ergodic GMMs account for the spatial distribution of ground motions due to path effects related to the 3-D velocity structure. For the ergodic GMM, the functional formulation of the natural log of FAS values, i.e., ln(Y), is given by

$$\ln(Y; M, R_{rup}, D_{hyp}) = \alpha_0 + \alpha_1 M + \alpha_2 \ln(R_{rup}) + \alpha_3 D_{hyp} + \delta B_e$$
$$+ \delta S2S_s + \frac{R_{rup}}{100} \delta P_e + \delta WS_{es}, \tag{9}$$

where $\alpha_1 M, \alpha_2 \ln(R_{rup}), \alpha_3 D_{hyp}$ denote magnitude scaling, geometrical spreading, and depth scaling terms, respectively. $\delta B_e, \delta S2S_s, \delta P_e$ represent the source, site, and path effects with zero-mean normal distributions. $\delta WS_{es}$ is the within-site residual, which is also assumed to be normally distributed. The coefficients are derived through linear regression, ensuring a smooth spectrum and imposing physical constraints on the coefficients[85]. Moreover, the total standard deviation of ergodic GMMs is defined as,

$$\sigma_{fas} = \sqrt{\tau^2 + \phi_{S2S}^2 + \phi_{SS}^2}, \tag{10}$$

where $\tau, \phi_{S2S}, \phi_{SS}$ denote the standard deviation of the mixed-effects coefficients $\delta B_e, \delta S2S_s, \delta WS_{es}$, respectively. The parameters of the ergodic GMM in the SFBA at different frequency bands (2, 5, 10, and 15 Hz) are detailed in Table 2.

For the non-ergodic GMMs, the source, site, and path terms are considered spatially dependent, which are functions of the coordinates of sources and sites. Therefore, the non-ergodic GMM is re-

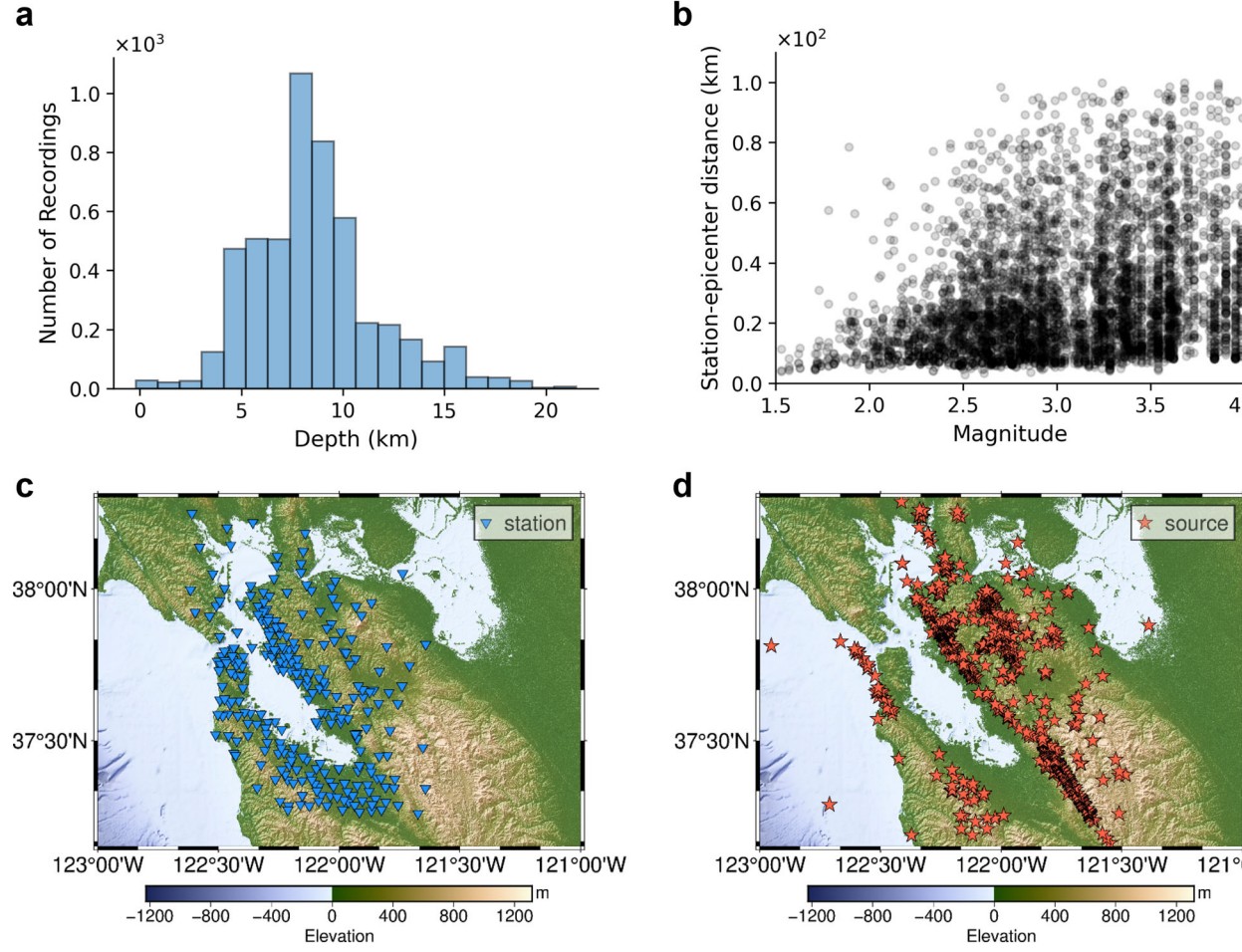

**Fig. 6 | An overview of the earthquake dataset of the H1 component in the San Francisco Bay Area. a** Shows the distribution of earthquake depths. **b** Presents the magnitude-distance distribution of this dataset. Each dot indicates the magnitude-distance of each source-receiver pair. **c, d** The spatial distributions of observation stations and earthquake sources, respectively.

## Table 2 | Summary of coefficients in the ergodic ground motion model at different frequencies

| Frequency | $\alpha_0$ | $\alpha_1$ | $\alpha_2$ | $\alpha_3$ | $\sigma_{fas}$ |
|---|---|---|---|---|---|
| 2 | −16.2739 | 3.0407 | −1.4842 | 0.0060 | 0.72 |
| 5 | −13.5466 | 2.6166 | −1.7052 | 0.0178 | 0.77 |
| 10 | −13.4538 | 2.2958 | −1.9021 | 0.0339 | 0.81 |
| 15 | −14.4238 | 2.1082 | −2.0292 | 0.0461 | 0.87 |

written as[49],

$$
\ln(Y; M, R_{rup}, D_{hyp}, \dots, \vec{te}_{es}, \vec{ts}_s) = LANNP26_{Adj-erg}(M, R_{rup}, D_{hyp}) + \delta L2L(\vec{te}_{es})
$$
$$
+ \delta S2S(\vec{ts}_s) + \delta P2P(\vec{te}_{es}, \vec{ts}_s) + \delta B_e^0 + \delta WSP_{es}
$$
(11)

where $LANNP26_{Adj-erg}$ is the median from the ergodic model as shown in Eq. (9), $\delta L2L(\vec{te}_{es})$, $\delta S2S$, $\delta P2P$ are the median shifts of the source, site, and path terms. $\vec{te}_{es}$ and $\vec{ts}_s$ represent the earthquake and site locations. Additionally, the term $\delta B^0 + \delta WSP_{es}$ denotes the aleatory variability apart from the systematic source, site, and path effects. The Gaussian Process (GP) regression is leveraged to fit the available ground motion data within the SFBA dataset by providing the medians and epistemic uncertainty[49]. The non-ergodic GMMs simulate FAS maps thanks to the capability of spatial interpolation of GP.

Furthermore, let the $\tau_0$ and $\phi_{SP}$ represent the standard deviations of $\delta B_e^0$ and $\delta WSP_{es}$, respectively. The total aleatory variance of non-ergodic GMMs is formulated as,

$$
\sigma_{NE}^2 = \tau_0^2 + \phi_{SP}^2.
$$
(12)

We use the same earthquake scenario and $100 \times 100$ spatial coordinates of stations to synthesize the non-ergodic FAS maps.

## Data availability

Waveform data, metadata, or data products for this study were accessed through the Northern California Earthquake Data Center (NCEDC) under DOI: 10.7932/NCEDC (https://doi.org/10.7932/NCEDC). The downloaded datasets were preprocessed and provided from[82] under https://doi.org/10.17603/ds2-necm-5q32.

## Code availability

All source code to reproduce the results in this study is available on GitHub at https://github.com/paulpuren/cgm-gm and is archived on Zenodo at https://doi.org/10.5281/zenodo.18480270[86]. We provide the training, generation, evaluation, and visualization details.

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

## Acknowledgements

This work was supported by the Laboratory Directed Research and Development Program of Lawrence Berkeley National Laboratory under U.S. Department of Energy Contract No. DE-AC02-05CH11231. M.W.M. and R.N. acknowledge the support by the Statewide California Earthquake Center (Awards Nos. 24123 and 25303). SCEC is funded by NSF Cooperative Agreement EAR-2225216 and USGS Cooperative Agreement G24AC00072-00. Additional support was provided by DOE Grant DE-SC0016520. M.W.M. would also like to acknowledge the DOE Competitive Portfolios grant and the DOE SciGPT grant. P.R. would like to thank Dr. Rasmus Malik Hoeegh Lindrup for his valuable discussions on generative modeling.

## Author contributions

P.R. developed the algorithm and performed the tests and analysis. P.R., R.N., and M.W.M. co-designed the study. P.R., I.N., and J.S. conducted the experiments. M.L. built the SFBA dataset and developed the ergodic and non-ergodic ground motion models. M.L., Z.B., and N.N. performed the geophysical evaluations. O.A.M., D.M., O.A., and N.B.E. contributed to the algorithm design and provided support for model development. R.N. and M.W.M. advised the research. All authors contributed to the research discussions, writing, and editing of the paper.

## Competing interests

The authors declare no competing interests.
