## [Transparent Peer Review file · Nature Communications]

Learning Earthquake Ground Motions via Conditional Generative Modeling

Corresponding Author: Dr Pu Ren

Version 0:

Reviewer comments:

Reviewer #1

(Remarks to the Author)

Review of manuscript of "Learning Physics for Unveiling Hidden Earthquake Ground Motions via Conditional Generative Modeling"

The manuscript presents the results of conditional generative models for ground motion simulation. The approach is based on a dynamic variational autoencoder, which does not have the drawbacks and difficulties of CGANs. The authors demonstrate promising results of this approach for scenarios with small magnitude in the San Francisco Bay area. The author investigated the potential of the proposed approach for capturing different characteristics of ground shaking.

I think the proposed approach and the results are interesting. In particular, the proposed method has a significant application in the improvement of shaking maps in early warning systems and seismic hazard analysis for regions with low magnitude. I believe this type of generative model is part of the next generation of ground motion modeling and simulations. I suggest publishing the manuscript after revision.

Here are my comments:

1) Page 4, Line 8: In signal processing, it is not common to use the Short-Term Fourier Transform. The STFT stands for the short-time Fourier transform.

2) I enjoyed Figure 04 and how the proposed method captures the arrival time. However, it is important to investigate the duration of ground shaking (e.g., significant duration) of ground-shaking signals between real and generated data. This investigation is very important for investigating shaking in geotechnical and seismic hazard analysis.

3) In the studied range of magnitude, the source is assumed to be a point source, and in this case, the hypocenter and rupture distances are the same. You mentioned R_{hyp} but consider it a rupture distance. It is better to be consistent and use one of them.

4) You considered R_{hyp} and depth together as conditions. I completely understand why depth is important, but this parameter is part of R_{hyp} , too. Why did you not use R_{epi} instead of R_{hyp} ?

5) Figure 02(d,e): The Y-axis limits for these two plots are not consistent, making it difficult to compare them.

6) Figure 03: The label of the colorbar for Error is missing ($\ln(\hat{FAS}/FAS)$).

7) The author did not explain why they consider epicenter-station azimuths as a condition parameter in the generative model. Does this parameter capture the anisotropy of media around the source and the path effect?

8) The author should mention that the proposed method and incorporated parameters can not capture the site effects.

9) Page 11: The author mentioned that using the time domain instead of the time-frequency domain can improve the generative model since it no longer requires phase retrieval. I agree with the intrinsic bias of phase retrieval because of the

ill-posedness of this problem. However, I think using the time domain directly is more challenging since the waveform is nonstationary and requires decomposing the signal into different frequencies. It is just my opinion.

10) I think the introduction and literature on ground motion simulations are incomplete. There are approaches that simulate ground shaking in the time-frequency domain based on a stochastic approach (Sabetta et al. 2021).

11) Figure 1, 6, 7, and 8: The time axes in simulated records are different, which is misleading. I suggest modifying it to the same maximum time for all seismic records or mentioning in the capture that the records are limited to a specific duration.

Reference:

Sabetta, F., Pugliese, A., Fiorentino, G. et al. Simulation of non-stationary stochastic ground motions based on recent Italian earthquakes. *Bull Earthquake Eng* 19, 3287–3315 (2021). <https://doi.org/10.1007/s10518-021-01077-1>

(Remarks on code availability)

I was curious about the code and the approach. The code does work and the results are reproducible.

Reviewer #2

(Remarks to the Author)

The authors present a Variational Autoencoder (VAE) referred to as conditional generative modeling framework for ground motion (CGM-GM) that learns sequences of short-term Fourier Transforms of earthquake time series, conditioned on the coordinates of the earthquake source and receivers, and other physical parameters such as the earthquake magnitude and hypocentral depth.

The authors argue, correctly, that the stochastic approach of generating ground motion time series is computationally efficient but lacks the underlying physics framework and leads to waveforms that do not represent realistic arrivals of the different seismic phases. On the other hand, physics-based simulations are physically sound but are computationally taxing and carry significant uncertainties in the representation of the source, path and nonlinear, heterogeneous site effects alike.

Machine learning approaches such as the model presented by the authors have emerged in the past 5 years, taking advantage of the advances in artificial intelligence algorithms and the proliferation of simulated and recorded ground motion data. The authors, however, use erroneous statements to demonstrate the need for their model compared to alternative machine learning methodologies for ground motion generation. Furthermore, their results do not support their claims on being able to simulate path effects and high frequency ground motions. I will elaborate below:

1. The authors claim that the model can generate [spatially continuous earthquake ground motion waveforms]; however, the methodology described in the text does not support this claim adequately. From the framework outlined, it is unclear how the model learns the spatial correlations between different stations for a given event. The inference process described suggests that the VAE generates 1D time histories independently for each station, and there is no detailed description of how a conditional 1D model can capture the complex 3D spatial-temporal functions inherent in earthquake data.

2. If the model is indeed learning the path effects, this should be demonstrated by evaluating multiple scenarios similar to the one in Figure 2c, and evaluating the uncertainty along paths where there are no data. In the Gaussian Process framework that non-ergodic models are built upon, lack of data translate into having the full epistemic uncertainty in the adjustment factors. If the uncertainty that the authors get is constant in areas with no data, then what they are learning is not path effects, but some averaged effects for a range of sources (and not the true wave propagation physics along the underlying medium).

3. The paper doesn't clearly explain how the VAE model generates synthetic ground motion samples. As far as I understand it, there are two ways of using VAE to generate time series:

3.1 The first method involves directly sampling from the latent space, which is typically assumed to be a standard Gaussian distribution, followed by passing these samples through the decoder. This approach aligns with the sampling processes used by other generative models such as GANs and Diffusion Models. If this is the case, then the model is at its core like the GAN models that the authors criticize in the introduction.

3.2 The second method involves reconstructing data from latent samples, where an input x (waveform sample) is first encoded to a latent representation z , followed by decoding z back to x . This method, however, does not allow for the generation of ground motion for arbitrary (future) earthquakes and sensor locations, as it relies on having existing data to reconstruct from. If this is the case, then the model is not able to generate time series at any arbitrary spatial location, let alone at spatially continuous wavefields.

4. Following up on 3.1, the paper dismisses the use of GAN citing that "GAN models cannot capture 3D wave propagation effects due to limited input conditional variables" is not a limitation of GANs. Rather, the ability of GANs to model complex phenomena like 3D wave propagation largely depends on the design of the input space and the conditioning of the model. The inclusion of more or appropriate conditional variables could potentially mitigate this limitation, suggesting that the model's design, rather than the model type, influences its capability. This is one of many erroneous statement examples

5. VAEs are known to generate samples of lower quality compared to GAN and Diffusion models. I cannot see the advantage of using the conditional VAE model compared to other-state-of-the-art generative models. While GAN and Diffusion models are shown to have universal approximation ability under mild conditions, the universal approximation ability of VAE is constrained by the reliance on the Gaussian assumption in the latent space. The training objective of VAE relies on KL divergence, which encourages the latent space to match a predefined distribution (typically Gaussian). This regularization can sometimes be too strong, leading to posterior collapse and not covering all modes of the data distribution. To circumvent issues with training stability of GANs, Diffusion models such as DDPM have significantly improved their training and inference speeds and can be viewed as an extension of VAEs that also mitigates earlier concerns about training costs. Lastly, the conditional VAE model relies on a phase retrieval method that has been shown to perform worse than learning waveforms directly in the time domain, similarly to the approach explored by GAN models.

6. The spike-like issues observed in Figures 6 and 7 appear to introduce high-frequency artifacts, which are not observed in previous machine-learning-based ground motion synthesis models. The authors should conduct an ablation study to determine whether these artifacts are attributable to the chosen model itself or are a result of the conditional variables used. This study could help clarify the source of the artifacts and guide improvements in the model design or variable selection. Knowing that results deviate from the median of GMMs and NGMMs in the high frequency regime, as evidenced also by the statistics in Figure 2 d and e, makes the generation of FAS maps at 10Hz irrelevant; the maps that would be useful to see are the <1Hz maps and their uncertainty, that show the capability of the model to capture path effects, if it does, as the authors claim.

7. The selection of baseline models for validation in this study appears to be inadequate. Fine-tuned GANO models such as the recently published version of [32] should be included in the comparison.

8. The model that is trained on small magnitude data, should also show the capability for reasonable scaling, with increased uncertainty, for larger magnitude events. If it doesn't, then there is no value in having a model like this for California.

(Remarks on code availability)

Reviewer #3

(Remarks to the Author)

The authors of the manuscript entitled "Learning Physics for Unveiling Hidden Earthquake Ground Motions via Conditional Generative Modeling" proposed a dynamic VAE, featured by variational sequential architectures (e.g., RNNs) in both prior and posterior distributions, which facilitates the capture of the temporal evolution (dynamics) of time series. Thanks to this Conditional Generative Modeling for Ground Motion (CGM-GM) strategy, the authors synthesize high-frequency and spatially continuous earthquake ground motion scenarios in the San Francisco Bay Area. Interestingly, CGM-GM embeds earthquake magnitude and hypocenter coordinates as well as sensors' geographic coordinates, so to reproduce the natural ground motion variability. CGM-GM was trained with small-magnitude data, which is remarkable because they challenge small Signal-to-Noise Ratios.

The manuscript is very well written. The presented results are convincing. The CGM-GM framework is effective and robust for generating earthquake ground motions. Its lightweight architecture is highly appreciated. The most interesting aspect of our proposed generative model lies in its claimed ability to approximate ground motions for arbitrary (future) earthquakes and sensor locations.

Some comments:

- In the Introduction, the authors overlooked some recent literature on earthquake ground motion generation using neural operators and graph neural networks. These represent a credible alternative to the CGM-GM presented in the manuscript and it would be interesting to highlight what are the advantages and limitations of adopting CGM-GM instead of one of these other approaches.

- What steps and data would the authors deem necessary to apply CGM-GM to another arbitrary region of the world? The title entices the readership but it seems that CGM-GM would be hardly adoptable in regions with poor seismic record and/or varying focal mechanisms. It is in this reviewer's opinion that the title should reflect the claim the authors make that their primary interest is in generating waveforms along the Hayward Fault, with no domain shift for the earthquake rupturing processes.

- Did the authors consider using ground motion numerical simulation to expand their training database? If so, how would they mix synthetics with records within the learning framework?

- As per this reviewer's understanding, each 3-component single-station time-history is encoded separately, along with its geographical coordinate. How is spatial correlation encoded into the latent space, if so? Peak values contour plots seem rather realistic, but what about the whole time series? This question arises from the fact that the phase spectra and the ground motion coherence are of paramount importance in earthquake engineering and seismic hazard modelling.

- The readership would benefit from a Figure showing extensive goodness-of-fit metrics comparisons. This reviewer suggests to adopt the modified Anderson's criteria, found in Olsen et al., 2010. Other GoF metrics would probably mislead the readership, because CGM-GM is conceived for stochastic generator of ground motions, not for historical ground motion reconstruction.

Minor comments and questions are found in the attached pdf.

(Remarks on code availability)

Data report is not available, nor data themselves. Testing failed because of the lack of test dataset.

I suggest to add some demonstrative notebook/tutorial.

Version 1:

Reviewer comments:

Reviewer #1

(Remarks to the Author)

I am satisfied with the revisions to the manuscript. I recommend this article for publication.

(Remarks on code availability)

Reviewer #2

(Remarks to the Author)

I would like to start by thanking the authors for putting effort to respond, to the extent that they have, to my questions, and improving the manuscript.

I still think that the work is valuable but its value is significantly overstated, to the point of even erroneously representing the contribution. I will summarize here, again, the main issues I have with this work. Some of the improvements that the authors have made have offered clarity while other improvements have reinforced false claims of this work. I insist that the authors should clarify the true contributions and remove all the comparisons and claims that point towards any other directions. Specifically:

1. The model that the authors are presenting here is one dimensional. This means that each event-station pair is trained independently of any other station, despite the fact that the coordinates of the source and station are introduced as conditional variables. There is no global variable for the VAE to determine that two adjacent stations with the same source coordinates, from the same event, come from the same earthquake. In this sense, the model IS NOT LEARNING wave propagation.
2. Since the model is not learning wave propagation, the title LEARNING PHYSICS is false. Its learning a mapping from one point to another, independent to each other, but no coherency; and since the medium from the source to the surface is fixed, the high frequencies (here 10 Hz) are learning some shallow crustal amplification functions which one could similarly capture by replacing the source-station coordinates with Vs30. Instead, the authors are trying to learn a 3D problem with a 1D model which, not only is impossible, but also leads to erroneously spiked strong motion records like the ones they show in Figure 7.
3. Another way to see what I am saying is to observe the incoherent pattern that emerges in the longer period range, at 2Hz for example, where 'path' and 'site' effects become muddled. There, Figure 14 shows that adjacent stations respond independently of each other for the same 'event'. In other words, there is no spatial continuity. The authors are just querying the model at adjacent stations, whose waveforms, however, even for the same location of source and magnitude of earthquake, will not necessarily be spatially correlated like they are originating from the same event.
4. In that sense, the best comparison the authors can make is against the ergodic GMMs in Nor Cal. These treat each GM time series separately with some conditional variables. The non-ergodic GMMs implement within event variability terms by studying the residuals relative to the ergodic terms of all the event terms at the same time across the strong motion network using GPs. Why dont they show in Figure 2 comparison against the ergodic GMMs? This is a much more fair comparison since the non-ergodic GMM model is a higher dimensional model than the cGM-GM. And definitely statements like 'outperforming a state-of the art non-ergodic GMM' should be deleted since they are misinforming the audience that the model is capturing effects that it is not.
5. Despite the fact that cGM-GM uses event and station coordinates, because the stations 'dont know' that their motions come from the same event, it cannot be used to plot scenaria. In that sense, all the scenaria plots like Figure 2b and 2c should be replaced with some median amplitude of many realizations, to be compared with median amplitudes of ergodic GMM plots.
6. Page 7, line 3 of the text: [... implicit assumption of a homogeneous Earth...] should be replaced with the [... implicit assumption of a radially homogeneous Earth...]. We should also note here that this is a limitation of the baseline model that

the authors selected; a more fair comparison would be to select a baseline model conditional on magnitude, distance, and a site response proxy (the same they used in their empirical GMM maps) in which case they would recover most of the capabilities that they are able to replicate with their cGM-GM model, possibly with a better time series synthesis capability.

7. Top of page 8: again erroneous statement: the agreement of the non-ergodic median here is purely phenomenological and if they plot the ergodic median the agreement will likely be better. Also, they should not be plotting scenarios, but median of scenarios, since their model is also ergodic (in the sense that it is learning independent pairs of source-stations and not within event variability like non-ergodic empirical models).

8. Page 10: bottom of first paragraph: Figure 4e shows bias in the arrival times, not a good correlation as the authors claim. The heat map is clearly off center.

9. Figure 5b and description of Figure on page 10: The SA trends at 0.1sec are completely off: the empirical model saturates at M8 due possibly to nonlinear site response and the cGM-GM increases almost exponential (due to the fictitious spikes we described above). The fact that the lines cross at some point doesn't mean there is agreement.

10. The 2Hz and 5Hz median predictions vs. the ergodic GMM maps should be plotted along with the 10Hz. Engineers don't care about frequencies as high as 10Hz nearly as much as they care about low frequencies.

(Remarks on code availability)

Reviewer #3

(Remarks to the Author)

The authors satisfactorily addressed almost all of the comments raised by this reviewer in the first round, except Comment 2. The sense of the question was if CGM-GM could be adopted zero-shot. As per the explanation given, the suspect is that the CGM-GM is not fit for zero-shot. A comment on this would be highly appreciated.

The transparency of showing the goodness of fit in Fig.10 is highly appreciated.

Filippo Gatti

(Remarks on code availability)

The authors uploaded the data report to the DesignSafe (see link: <https://www.designsafe-ci.org/data/browser/public/designsafe.storage.published/PRJ4573>). The data report is also updated in the revised manuscript (see Page 20). Additionally, they have included tutorial examples, including guidance on training models and performing evaluations from various perspectives. Please see the updated materials in our GitHub repository: <https://github.com/paulpuren/cgm-gm>. The latter github repository contains a README.md and two notebooks to run the application.

Version 2:

Reviewer comments:

Reviewer #2

(Remarks to the Author)

I would like to thank the authors for addressing my comments to some extent. There still are unresolved concerns in the revised version that I need to see addressed, however. I want to make sure that the authors properly highlight the contributions of the manuscript, which are important, without misleading the earthquake engineering audience who may be interested in using the results of their work.

For this purpose, I use sections of the text to point out inconsistencies between what the authors's model is and what they say it is in the text (line numbers would have been helpful in that sense but I'll use page numbers and approximate locations):

1. p3 "Our results demonstrate the excellent performance of our proposed model in learning the underlying wave propagation characteristics..." There is no evidence, nor it is expected to be any evidence of the model to learn wave propagation: for that, the authors would have to show spatiotemporal evolution of the wavefield which they don't because this is not what the model is supposed to be doing. All references to how great a job the model does in learning wave propagation should be removed. They are false. The model is learning some form of frequency response median from irregularly spaced stations and that is great. Say that.

2. p3. "Our model learns a stochastic mapping that captures wave propagation characteristics between earthquake sources and stations..." again, rephrase the misleading statement.

3. p4. "Nevertheless, we demonstrate that the model is capable of capturing spatial continuity in the median generated ground motions." No you don't. You demonstrate that the model is capable of capturing spatial continuity in the median generated FAS.

4. p6. "We utilize an existing earthquake event that occurred at 3:16 AM on October 21, 2011." This is misleading because, as mentioned earlier, the model is not designed to capture scenarios but median frequency responses. The authors should make it explicit that the model cannot capture event level spatiotemporal variability.

5. p13. "The primary objective of our proposed method is to generate spatially continuous and physically meaningful ground motions. Our model learns a stochastic mapping that captures wave propagation characteristics through a combined representation of source, site, and path effects." Again, rephrase the capturing of wave propagation. Call it frequency response or something that explicitly takes out the spatiotemporal continuity misinterpretation that runs across the manuscript.

6. The abstract needs to be rephrased accordingly.

Other comments:

7. Why compare the scaling of large magnitude events to ASK14 and not to the non-ergodic GMM for California by Lavrentiadis et al (2023)? It seems to be a much more appropriate comparison given the flow of your work than switching metric and suddenly comparing to the Sa of ASK14.

(Remarks on code availability)

Version 3:

Reviewer comments:

Reviewer #2

(Remarks to the Author)

Again, I would like to thank the authors for taking the time to revise their manuscript and bring the manuscript content closer to what their work actually is about.

In their eagerness to achieve more, however, they --again-- have introduced incorrect representations of the material presented. I am baffled by this attitude, but I do believe that this group of young computer scientists with fresh ideas should gain visibility in the community which is why I continue to suggest revisions and do not recommend that the manuscript is rejected. Here are some more items for the authors to reconsider:

1. The title and abstract need revision. The word 'physics' is misleading in the title and now the abstract is also misleading:

[Predicting high-fidelity ground motions for future earthquakes is crucial for seismic hazard assessment and infrastructure resilience. Conventional empirical simulations suffer from sparse sensor distribution and geographically localized earthquake locations, while physics-based methods are computationally intensive and require accurate representations of Earth structures and earthquake sources.  We propose a novel artificial intelligence (AI) waveform generator, Conditional Generative Modeling for Ground Motion (CGM-GM). CGM-GM leverages earthquake magnitudes and geographic coordinates of earthquakes and sensors as inputs, capturing spatially continuous Fourier amplitude spectra (FAS) as well as properties such as P and S arrivals, and waveform durations without explicit physics constraints....]

The abstract has been edited by thereby lies the problem: the authors are not proposing a waveform generator. They are proposing an FAS median mapping tool that, if fed with a white noise phase, can produce random phase waveforms, but they are not physically correct! This is not acceptable at least on my end.

2. I appreciate that the authors went ahead and incorporated the LA21 NGMM for larger events in Figure 5. Where are the continuous changes in slope in the magnitude scaling coming from? The implementation of the NGMM is clearly incorrect and I am not sure where the error is by looking at the figure.

3. Furthermore, the trend of the cGM-GM and the NGMM is opposite. The proposed model doesn't saturate at large magnitude events, on the contrary -- it explodes, yet another piece of evidence that it doesn't really learn physics. The authors should think really carefully of what this figure represents.

4. The same is evident in the high frequency regime of the small events. The model is learning a lot of noise, which is why it is predicting --at best-- median amplitudes up to 11Hz. Everything above that, including the trend, should be commented on in the text, since it shows that the model doesn't extrapolate well.

(Remarks on code availability)

N/A

Version 4:

Reviewer comments:

Reviewer #2

(Remarks to the Author)

I am satisfied with the changes and thank the authors for sticking with me through several revisions of their manuscript. I do not have any further comments on this manuscript.

(Remarks on code availability)

N/A

Learning Physics for Unveiling Hidden Earthquake Ground Motions via Conditional Generative Modeling

NCOMMS-24-45416

Pu Ren, Rie Nakata, Maxime Lacour, Ilan Naiman, Nori Nakata, Jialin Song, Zhengfa Bi, Osman Asif Malik, Dmitriy Morozov, Omri Azencot, N. Benjamin Erichson, and Michael W. Mahoney

Dear Editor and Reviewers,

We sincerely appreciate the insightful and constructive comments from the reviewers, which have helped improve our manuscript. We are pleased that all reviewers acknowledge the significance of our research and see the merit of our manuscript. In particular, we thank the reviewers for recognizing the effectiveness (Reviewers 1 and 3), robustness (Reviewer 3), and computational efficiency (Reviewers 2 and 3) of our proposed method in synthesizing arbitrary small-magnitude earthquakes. We extend our gratitude to Reviewer 3 for commending the overall quality of the manuscript.

Our point-by-point responses are provided below marked in blue and the corresponding manuscript revisions are highlighted in red. Additionally, we have updated all figures in the revised manuscript, as we made minor adjustments to the training hyperparameters to ensure optimal network performance. These modifications did not change the interpretation of the results. We hope our manuscript will be found worthy of publication in this revised form.

I. RESPONSE TO REVIEWER 1

General comment: The manuscript presents the results of conditional generative models for ground motion simulation. The approach is based on a dynamic variational autoencoder, which does not have the drawbacks and difficulties of CGANs. The authors demonstrate promising results of this approach for scenarios with small magnitude in the San Francisco Bay area. The author investigated the potential of the proposed approach for capturing different characteristics of ground shaking.

I think the proposed approach and the results are interesting. In particular, the proposed method has a significant application in the improvement of shaking maps in early warning systems and seismic hazard analysis for regions with low magnitude. I believe this type of generative model is part of the next generation of ground motion modeling and simulations. I suggest publishing the manuscript after revision.

Reply: Thank you very much for your positive feedback and recommendation for publishing after revision. We will address your concerns point-by-point as follows.

Comment 1: Page 4, Line 8: In signal processing, it is not common to use the Short-Term Fourier Transform. The STFT stands for the short-time Fourier transform.

Reply: Thank you for pointing this out. We have updated the terminology to the commonly-used "short-time Fourier transform", as suggested. The modifications can be found on Page 4 of the revised manuscript.

Comment 2: I enjoyed Figure 04 and how the proposed method captures the arrival time. However, it is important to investigate the duration of ground shaking (e.g., significant duration) of ground-shaking signals between real and generated data. This investigation is very important for investigating shaking in geotechnical and seismic hazard analysis.

Reply: We agree with the reviewer that the shaking duration is also a critical factor in evaluating the simulated ground motions. We have included additional evaluations of the shaking durations for all simulated ground motion waveforms, as presented in Figure 4(e) of the revised manuscript (Page 11). This figure provides cross-validations of the shaking durations in the EW direction, demonstrating a good agreement between the generated samples and the ground truth. The duration in the NS direction shows a similar trend. The corresponding discussions have been added in Section *P and S arrival times and durations* (see Page 10).

FIG. 1. The cross-validation of the shaking durations in the EW direction. The density is normalized between 0 and 1.

Comment 3: In the studied range of magnitude, the source is assumed to be a point source, and in this case, the hypocenter and rupture distances are the same. You mentioned R_{hyp} but consider it a rupture distance. It is better to be consistent and use one of them.

Reply: Thank you for this suggestion. You are correct that the source is treated as a point source within the studied magnitude range. We have revised R_{hyp} to R_{rup} to represent the rupture distance throughout the paper. Meanwhile, please also see the updates in Figures 1e, 3a, and 7–12.

Comment 4: You considered R_{hyp} and depth together as conditions. I completely understand why depth is important, but this parameter is part of R_{hyp} , too. Why did you not use R_{epi} instead of R_{hyp} ?

Reply: We agree that depth is inherently included in R_{hyp} . To ensure clarity and consistency, we have replaced “rupture distance” with “epicentral distance” as one of the conditional variables in the CGM-baseline model (see Page 6). We have also updated the results with the latest setup in the revised manuscript (see Figure 2).

Comment 5: Figure 02(d,e): The Y-axis limits for these two plots are not consistent, making it difficult to compare them.

Reply: Thank you for bringing this to our attention. We have adjusted the Y-axis limits in Figure 2(d) and (e) to ensure consistency. The modifications can be found on Page 7 of the revised manuscript.

Comment 6: Figure 03: The label of the colorbar for Error is missing ($\ln(\text{FAS}/\hat{\text{FAS}})$).

Reply: Thanks for pointing it out. We have included the label of the colorbar for Figure 3(b). Please see the modifications on Page 9.

Comment 7: The author did not explain why they consider epicenter-station azimuths as a condition parameter in the generative model. Does this parameter capture the anisotropy of media around the source and the path effect?

Reply: Thank you for this question. First of all, we would like to clarify that epicenter-station azimuths were not included as a conditional parameter in our generative model. In the CGM-GM framework, we utilize earthquake magnitude, earthquake depth, and the geospatial coordinates of earthquake sources and seismic stations as conditional variables. For comparison, the CGM-baseline employs earthquake magnitudes, source depths, and epicentral distances as conditional variables. The details of conditional variables for CGM-GM and CGM-baseline are listed in TABLE (I). Our coordinate-based parameters allow us to effectively capture underlying heterogeneity. While we do not have evidence of strong anisotropy in the SFBA (for example, the USGS San Francisco Bay region 3D seismic velocity model (SFVM) does not include anisotropy, but reasonably explains observed waveforms), our conditional variable setup allows us to capture anisotropy.

Secondly, we would like to elaborate on the rationale behind using two different sets of conditional variables. The variables selected for the CGM-baseline model represent primarily 1D phenomena, whereas those for the CGM-GM model reflect 3D wave physics and subsurface properties, enabling a more comprehensive representation of seismic events. We have included an additional introduction in Section *Embedding conditional variables* (see Page 17).

TABLE I. Conditional variables used in CGM models.

Model	Conditional Variables
CGM-baseline	earthquake magnitudes, source depths, and epicentral distances
CGM-GM	earthquake magnitudes, geospatial coordinates of earthquake sources and stations

Comment 8: The author should mention that the proposed method and incorporated parameters can not capture the site effects.

Reply: Thanks for this suggestion. We would like to clarify that the primary objective of our proposed method is to generate spatially continuous waveforms. Our model captures wave physics by implicitly learning a combination of source, site, and path effects within the conditional generative AI framework. We didn't explicitly learn these effects individually. The site effect is regarded as a part of this learned representation. While our current framework does not separate site effects as independent components, addressing this limitation and developing methods for their explicit characterization will be an important direction for future research. We have revised the relevant sections in the updated manuscript to clarify this point. Please refer to Section *DISCUSSION* on Pages 12-13.

Comment 9: Page 11: The author mentioned that using the time domain instead of the time-frequency domain can improve the generative model since it no longer requires phase retrieval. I agree with the intrinsic bias of phase retrieval because of the ill-posedness of this problem. However, I think using the time domain directly is more challenging since the waveform is non-stationary and requires decomposing the signal into different frequencies. It is just my opinion.

Reply: This is an insightful comment. We agree with the reviewer that directly modeling waveforms in the time domain presents significant challenges, particularly in capturing long-range dependencies in time series data. For instance, recurrent neural networks often encounter vanishing and exploding gradient problems, which hinder their ability to learn complex patterns and long-term dependencies [1]. However, there is still some prior work in machine learning that successfully handles audio waveforms [2] and long-sequence time series data [3] in the time domain. Thus, while modeling directly in the time domain is indeed challenging, we believe it represents a worthwhile direction for further exploration.

References:

- [1] Naiman, Ilan, et al. "Utilizing Image Transforms and Diffusion Models for Generative Modeling of Short and Long Time Series." in *The Thirty-Eighth Annual Conference on Neural Information Processing Systems*, 2024.
- [2] Van Den Oord, Aaron, et al. "Wavenet: A generative model for raw audio." in *The 9th ISCA Speech Synthesis Workshop*, 2016.
- [3] Zhou, Linqi, et al. "Deep latent state space models for time-series generation." in *International Conference on Machine Learning*, 2023.

Comment 10: I think the introduction and literature on ground motion simulations are incomplete. There are approaches that simulate ground shaking in the time-frequency domain based on a stochastic approach (Sabetta et al. 2021).

Reference: Sabetta, F., Pugliese, A., Fiorentino, G. et al. Simulation of non-stationary stochastic ground motions based on recent Italian earthquakes. *Bull Earthquake Eng* 19, 3287–3315 (2021). <https://doi.org/10.1007/s10518-021-01077-1>

Reply: Thank you for recommending this paper. We agree that it provides valuable insights and will enrich the current literature review on stochastic methods for simulating ground motions. Accordingly, we have included this paper in the revised manuscript. Please see Page 2 (reference [4]) for the updated citation.

Comment 11: Figure 1, 6, 7, and 8: The time axes in simulated records are different, which is misleading. I suggest modifying it to the same maximum time for all seismic records or mentioning in the capture that the records are limited to a specific duration.

Reply: Thank you for this comment. In the figures mentioned, the time axes reflect the motion duration for each seismic record, which may vary depending on the specific event. We aim to highlight the most relevant portions of the waveforms for analysis. However, we acknowledge that this could potentially cause misunderstanding. To address this, we have added clarifications in the figure captions (see Pages 5 and 33), specifying that the displayed waveforms are limited to their respective motion durations.

Comment 12: Remarks on code availability: I was curious about the code and the approach. The code does work and the results are reproducible.

Reply: Thank you for reviewing our code. We have added additional tutorial examples, including guidance on training models and performing evaluations from various perspectives. Please see the updated materials in our GitHub repository: <https://github.com/paulpuren/cgm-gm>.

II. RESPONSE TO REVIEWER 2

General comment: The authors present a Variational Autoencoder (VAE) referred to as conditional generative modeling framework for ground motion (CGM-GM) that learns sequences of short-term Fourier Transforms of earthquake time series, conditioned on the coordinates of the earthquake source and receivers, and other physical parameters such as the earthquake magnitude and hypocentral depth.

The authors argue, correctly, that the stochastic approach of generating ground motion time series is computationally efficient but lacks the underlying physics framework and leads to waveforms that do not represent realistic arrivals of the different seismic phases. On the other hand, physics-based simulations are physically sound but are computationally taxing and carry significant uncertainties in the representation of the source, path and nonlinear, heterogeneous site effects alike.

Machine learning approaches such as the model presented by the authors have emerged in the past 5 years, taking advantage of the advances in artificial intelligence algorithms and the proliferation of simulated and recorded ground motion data. The authors, however, use erroneous statements to demonstrate the need for their model compared to alternative machine learning methodologies for ground motion generation. Furthermore, their results do not support their claims on being able to simulate path effects and high frequency ground motions.

Reply: Thank you for your thoughtful feedback. We appreciate the reviewer’s recognition of our general arguments and the context of our proposed approach. We understand that certain aspects of the original manuscript may have led to some misunderstandings. We provide a detailed point-by-point response below to address your concerns.

Comment 1: The authors claim that the model can generate [spatially continuous earthquake ground motion waveforms]; however, the methodology described in the text does not support this claim adequately. From the framework outlined, it is unclear how the model learns the spatial correlations between different stations for a given event. The inference process described suggests that the VAE generates 1D time histories independently for each station, and there is no detailed description of how a conditional 1D model can capture the complex 3D spatial-temporal functions inherent in earthquake data.

Reply: Thank you for raising this important point. To address your concerns, we would like to clarify how our CGM-GM framework captures spatial correlations and generates spatially continuous earthquake ground motion waveforms from two perspectives.

Capturing 3D spatiotemporal patterns: While the generated waveforms are 1D time series, the learning process is inherently 3D due to the conditional variables embedded in the model. The embedding

module incorporates the 3D setup of geospatial coordinates, including longitude, latitude, and depth of the source and site locations. This allows our CGM-GM model to learn and capture the complex spatiotemporal patterns of earthquake data. To be more concrete, let us consider a general case of predicting seismic waveforms at a station x_r generated from a source x_s , which is given by

$$\mathcal{F}(t, x_s, x_r, m) = \mathcal{S}(t, x_s) * \mathcal{R}(t, x_r) * \mathcal{G}(t, x_s, x_r), \quad (1)$$

where $\mathcal{F}(\cdot)$ denotes the prediction function that generates waveforms. $\mathcal{S}(\cdot)$, $\mathcal{R}(\cdot)$, and $\mathcal{G}(\cdot)$ represent the source effects, site effects, and Green’s function (or path effects), respectively. $*$ is the convolution. In our conditional generative modeling framework, this function $\mathcal{F}(\cdot)$ is approximated by the neural networks. Specifically, we incorporate physical variables (i.e., earthquake magnitudes, depths, source and receiver locations) as conditional variables v . In our implementation, x_s represents source parameters, including magnitudes, source locations, and depths. The CGM-GM framework incorporates source effects, such as source mechanisms and their coupling with near-source geology. Note that this leads to implicitly assuming a specific source mechanism at each source location, which is reasonable. The conditional variable of a receiver location corresponds to x_r and we incorporate the site effect $\mathcal{R}(x_r, t)$. We do not separately model $\mathcal{S}(\cdot)$, $\mathcal{R}(\cdot)$ and $\mathcal{G}(\cdot)$, and hence these effects are learned as a combination through the neural network. We interpret that using many earthquake events allows us to learn $\mathcal{R}(\cdot)$ at a specific location, and using many stations lets us learn $\mathcal{S}(\cdot)$ for a specific event. Similarly, by using many earthquake-station pairs, we learn Green’s function $\mathcal{G}(\cdot)$. We have included additional clarifications in the revised manuscript. Please see Section *A.1 Details of conditional variables* on Pages 30-31.

Spatial continuity: As shown in Figure 4 of the revised manuscript (Page 11), our generated 1D time series exhibit waveform characteristics, including arrival times, peak ground velocities (PGVs), and durations. They closely match the observations across numerous realizations. Furthermore, we demonstrate the reproduction of key features, such as P and S arrival times through cross-plots of their values. These results indicate that 3D wave propagation, source, and site effects are properly captured in a non-ergodic manner. The spatial continuity of ground motions generated by our model is demonstrated in the FAS map at 10 Hz, as shown in Figure 2(c). The generated FAS map exhibits spatially continuous patterns, even between station pairs separated by significant distances (e.g., in the southeast of the map). This continuity reflects the capacity of our CGM-GM framework to capture underlying spatial heterogeneities and wave physics inherent in earthquake ground motions. However, we acknowledge that the continuity of details of waveforms (reflections and scatterings) is still under investigation and subject of the future study. A detailed discussion on this aspect is provided in Section *Waveforms and spatial continuity*. Please see Pages 6-8 of the revised manuscript.

Comment 2: If the model is indeed learning the path effects, this should be demonstrated by evaluating multiple scenarios similar to the one in Figure 2c, and evaluating the uncertainty along paths where there are no data. In the Gaussian Process framework that non-ergodic models are build upon, lack of data translate into having the full epistemic uncertainty in the adjustment factors. If the uncertainty that the authors get is constant in areas with no data, then what they are learning is not path effects, but some averaged effects for a range of sources (and not the true wave propagation physics along the underlying medium).

Reply: First of all, we would like to clarify that our proposed model captures a combination of site, source, and path effects within a conditional generative framework. The main goal of our approach is to generate spatially continuous ground motions rather than explicitly learning these effects as separate components. To further illustrate this, we present a pseudo-path effect, defined as the residual between

the predicted FAS map and the summation of the ergodic effect, site effect, and source effect obtained from the non-ergodic Ground Motion Model (GMM). The comparison of path effect from non-ergodic GMM and pseudo path effect from CGM-GM is presented in Fig. 2. The results indicate that the pseudo-path effect derived from our model generally aligns with predictions from the non-ergodic GMM. We recognize the significance of further disentangling site, source, and path effects and will explore this in future work. We have included more clarifications on this aspect in the revised manuscript. Please see Section *DISCUSSION* on Page 13.

Furthermore, we emphasize that our goal is not to replace the Gaussian Process framework used in non-ergodic GMMs with generative modeling. Instead, we view generative modeling as a complementary approach that can provide additional insights and serve as a promising tool alongside existing GMM methodologies.

The CGM-GM model does not separate epistemic and aleatory uncertainties. One of our recent studies [1] showed that the ground motion variability depends on the hyperparameter settings. We consider that exploring uncertainties is a necessary future direction and warrants a thorough investigation, considering the importance of epistemic and aleatory uncertainties in seismic hazard analysis. Our additional tests conducted for this revision (magnitude scaling for Comment 8 from Reviewer 2 and limited data for Comment 2 from Reviewer 3) provide important initial insights into uncertainties of the CGM-GM. First, the comparisons with the GMMs at the available stations show that the aleatory variability of our model matches that of the ergodic GMM. Second, the increase in the range of predictions with decreasing training data shows that epistemic uncertainty is captured in the CGM-GM. These results are an initial path to future works on uncertainty quantification that will make the CGM-GM more practical and important for future seismic hazard analysis. We have added more clarifications in Section *Waveforms and spatial continuity* of the revised manuscript (Pages 7-8).

FIG. 2. A comparison of path effect from non-ergodic GMM and pseudo-path effect from CGM-GM.

References:

[1] Nakata, Rie, et al. "Simulating seismic wavefields using generative artificial intelligence." *The Leading Edge* 44.2 (2025): 123-132.

Comment 3: The paper doesn't clearly explain how the VAE model generates synthetic ground motion samples. As far as I understand it, there are two ways of using VAE to generate time series:

- The first method involves directly sampling from the latent space, which is typically assumed to be a standard Gaussian distribution, followed by passing these samples through the decoder. This approach aligns with the sampling processes used by other generative models such as GANs and Diffusion Models. If this is the case, then the model is at its core like the GAN models that the authors criticize in the introduction.
- The second method involves reconstructing data from latent samples, where an input x (waveform sample) is first encoded to a latent representation z , followed by decoding z back to x . This method, however, does not allow for the generation of ground motion for arbitrary (future) earthquakes and sensor locations, as it relies on having existing data to reconstruct from. If this is the case, then the model is not able to generate time series at any arbitrary spatial location, let alone at spatially continuous wavefields.

Reply: First of all, we would like to clarify that our intention was not to criticize the application of GANs to earthquake ground motion generation but rather to mention the inherent challenges in their training processes. We acknowledge that GANs have achieved significant success in seismic applications but their training often requires tricks to address instability issues [1]. VAEs offer a more stable training framework and are relatively more accessible to scientists without extensive deep learning expertise, and they complement previous generative approaches. VAEs provide a straightforward implementation while retaining the ability to capture complex patterns through their probabilistic framework. This aligns with one of our goals of making generative modeling methods more approachable to the broader scientific community. To avoid confusion, we have rephrased this part and added more clarifications in Section *INTRODUCTION* (see Page 3).

Secondly, we provide clarification on the training and inference processes of the general VAE framework.

Training Stage (Reconstruction): This corresponds to the “second method” mentioned in this reviewer’s comment. During training, the VAE is optimized to reconstruct input waveforms from their encoded latent representations. Specifically, a given input x (e.g., a seismic waveform sample) is encoded into a latent variable z , and the decoder reconstructs x from z . This process enables the latent space to learn a meaningful representation of the underlying seismic data. Additionally, we define a prior distribution specifically for seismic data. While a standard Gaussian prior is commonly used, we employ a sequential Gaussian prior, which better aligns with the temporal dynamics inherent in seismic waveforms. The prior distribution acts as a regularization term through the KL divergence, enforcing the latent space distribution to remain close to the prior. The reconstruction loss and the KL divergence regularization guide the model in learning a smooth and structured latent space and facilitate effective sampling during the generative process. Meanwhile, by incorporating conditional variables (e.g., geospatial coordinates, earthquake magnitudes, and depths) into the model prior and the decoder, the entire framework ensures the generation of consistent and realistic ground motion waveforms for arbitrary earthquake scenarios.

Inference Stage (Generation): Unlike the training stage (reconstruction), the inference process (generation) does not rely on existing input waveforms x to synthesize ground motion data. During inference, ground motion waveforms are generated by directly sampling from the prior distribution and passing it through the decoder to produce new waveforms. The conditional variables are integrated simultaneously to ensure that the generated waveforms are physically meaningful. This part aligns with the “first method” mentioned in this reviewer’s comment.

The training and inference processes of VAEs are illustrated in Figure 1(b-c) and introduced in the *Dynamic VAEs* and *Network design* sections. We have also included more clarifications in the revised manuscript. Please see Page 16.

References:

[1] Arjovsky, Martin, Soumith Chintala, and Léon Bottou. "Wasserstein generative adversarial networks." *International conference on machine learning*. PMLR, 2017.

Comment 4: Following up on 3.1, the paper dismisses the use of GAN citing that "GAN models cannot capture 3D wave propagation effects due to limited input conditional variables" is not a limitation of GANs. Rather, the ability of GANs to model complex phenomena like 3D wave propagation largely depends on the design of the input space and the conditioning of the model. The inclusion of more or appropriate conditional variables could potentially mitigate this limitation, suggesting that the model's design, rather than the model type, influences its capability. This is one of many erroneous statement examples

Reply: We would like to reiterate that it was not our intention to dismiss the use of GANs for ground motion generation (as noted in our response to Comment 3). In the original manuscript, we aimed to mention that previous applications of GANs in this context have often employed a different set of conditional variables, which may limit their ability to model 3D wave propagation effects comprehensively. We agree with the reviewer that the ability of GANs, or any generative model, to capture complex phenomena could be improved by including appropriate and sufficient conditional variables.

To address any potential misunderstanding, we have revised the relevant sections in the manuscript to clarify our position. We now emphasize that the limitations mentioned are tied to the specific design choices in prior work, rather than the GAN framework itself. Please see Page 3.

Comment 5: VAEs are known to generate samples of lower quality compared to GAN and Diffusion models. I cannot see the advantage of using the conditional VAE model compared to other-state-of-the-art generative models. While GAN and Diffusion models are shown to have universal approximation ability under mild conditions, the universal approximation ability of VAE is constrained by the reliance on the Gaussian assumption in the latent space. The training objective of VAE relies on KL divergence, which encourages the latent space to match a predefined distribution (typically Gaussian). This regularization can sometimes be too strong, leading to posterior collapse and not covering all modes of the data distribution. To circumvent issues with training stability of GANs, Diffusion models such as DDPM have significantly improved their training and inference speeds and can be viewed as an extension of VAEs that also mitigates earlier concerns about training costs. Lastly, the conditional VAE model relies on a phase retrieval method that has been shown to perform worse than learning waveforms directly in the time domain, similarly to the approach explored by GAN models.

Reply: Thank you for your comment. We would like to provide clarifications from three perspectives: the choice of dynamic VAEs, the research focus of our paper, and the use of phase retrieval.

The choice of dynamic VAEs: We agree that vanilla VAE models can sometimes produce lower-quality samples compared to GANs or diffusion models. However, in our paper, we employ a dynamic VAE framework, which has been demonstrated to perform well for time series generation (see this recent work [1]). In this context, the dynamic VAE framework has shown competitive results compared to models such as TimeGAN [2]. Moreover, VAEs are lightweight, computationally efficient, and straightforward to implement, making them especially accessible for a wider range of domain scientists who may not have extensive experience with generative models. Due to these benefits, VAEs remain an active area of research in scientific machine learning [3,4,5].

Each generative model has distinct strengths and limitations. For example, GANs are often challenged by training instability, while Diffusion models generally require significant computational resources. To provide future clarity, we have added a detailed comparison of commonly used generative models in Section *A.2 Comparisons of generative models*. Please see Page 31. For your convenience, we also present the summary in Table II.

TABLE II. Summary of GenAI models in terms of generation fidelity, sampling efficiency, training stability, and compression (✓: Good, ✗: Suboptimal).

Model	High fidelity	Fast sampling	Training stability	Compression
VAE	✗	✓	✓	✓
GAN	✓	✓	✗	✗
Diffusion model	✓	✗	✓	✗

The research focus of this paper: We would like to emphasize that the primary focus of this work is not on the generative model itself but more on the ability to learn spatially continuous ground motions. VAEs provide a suitable and accessible framework for this purpose. While GANs and Diffusion models offer their own strengths, our choice of a dynamic VAE was motivated by its stability, ease of use, and effectiveness for time series learning [1]. Additionally, we are actively exploring Diffusion models in follow-up work (see Bi. *et al.* [6]) to address some of the limitations mentioned.

The use of phase retrieval: We recognize that phase retrieval methods can introduce numerical errors. However, directly learning waveforms in the time domain also poses significant challenges, as also noted by Reviewer 1. There is always a tradeoff between computational difficulty and model performance. Additionally, we have included the discussion of phase retrieval methods in Section *DISCUSSION* and pointed out that we will explore learning in the time domain as a future work. It is worth noting that many recent studies [5,7] have successfully used phase retrieval methods in generative modeling, validating its practicality in the context of ground motion generation.

References:

[1] Naiman, Ilan, et al. "Generative modeling of regular and irregular time series data via Koopman VAEs." in *The Twelfth International Conference on Learning Representations*, 2024.

[2] Yoon, Jinsung, Daniel Jarrett, and Mihaela Van der Schaar. "Time-series generative adversarial networks." *Advances in neural information processing systems* 32 (2019).

[3] Eckmann, Peter, et al. "Limo: Latent inceptionism for targeted molecule generation." *Proceedings of machine learning research* 162 (2022): 5777.

[4] Solera-Rico, Alberto, et al. " β -Variational autoencoders and transformers for reduced-order modelling of fluid flows." *Nature Communications* 15.1 (2024): 1361.

[5] Ning, Chunxiao, and Yazhou Xie. "Convolutional variational autoencoder for ground motion classification and generation toward efficient seismic fragility assessment." *Computer-Aided Civil and Infrastructure Engineering* 39.2 (2024): 165-185.

[6] Bi, Zhengfa, et al. "Advancing data-driven broadband seismic wavefield simulation with multi-conditional diffusion model", arXiv.

[7] Esfahani, Reza DD, et al. "TFCGAN: Nonstationary Ground-Motion Simulation in the Time–Frequency Domain Using Conditional Generative Adversarial Network (CGAN) and Phase Retrieval Methods." *Bulletin of the Seismological Society of America* 113.1 (2023): 453-467.

Comment 6: The spike-like issues observed in Figures 6 and 7 appear to introduce high-frequency artifacts, which are not observed in previous machine-learning-based ground motion synthesis models. The authors should conduct an ablation study to determine whether these artifacts are attributable to the chosen model itself or are a result of the conditional variables used. This study could help clarify the source of the artifacts and guide improvements in the model design or variable selection. Knowing that results deviate from the median of GMMs and NGMMs in the high frequency regime, as evidenced also by the statistics in Figure 2 d and e, makes the generation of FAS maps at 10Hz irrelevant; the maps that would be useful to see are the $< 1\text{Hz}$ maps and their uncertainty, that show the capability of the model to capture path effects, if it does, as the authors claim.

Reply: We have examined the waveforms generated by both the CGM-baseline and CGM-GM models and found that the spike-like artifacts appear in both cases. One of the representative examples is shown in Fig. 3. Our analysis suggests that these artifacts are primarily due to the phase retrieval methods used during waveform reconstruction. Since phase retrieval estimates phase information indirectly from amplitude information, it can introduce inaccuracies that may distort waveform shapes, particularly in the high-frequency range. However, we emphasize that these artifacts do not significantly impact the model’s ability to generate realistic Fourier Amplitude Spectra (FAS) maps, as the overall spectral characteristics remain well-preserved. Hence, comparing the FAS at 10 Hz, where the CGM-GM demonstrates the best fit to the observations, is reasonable.

FIG. 3. A representative example of comparing waveforms from CGM-GM and CGM-baseline.

We agree that evaluating lower-frequency FAS maps and their associated uncertainties would provide additional insights for path effects. However, the frequency range used in this study is limited to $[2, 15]$ Hz, and hence we cannot compute the FAS map at 1 Hz without building a model using wider frequency data. We selected $[2, 15]$ Hz to satisfy the S/N ratio over an entire frequency range at all used stations. Small-magnitude earthquakes typically exhibit very low S/N ratios at lower frequencies, which would significantly reduce the size and reliability of the training dataset. Nevertheless, extending the CGM-GM framework to incorporate lower-frequency data remains an important direction for future work. This could be achieved by fine-tuning the model using stations with high-quality low-frequency waveforms or by developing a framework and training strategies that effectively handle varying data distributions across different frequency ranges. For the discussion of uncertainty, please refer to our response to Comment 2.

Additionally, we present the FAS map at 5 Hz in Section *B.6 FAS Maps* (Pages 34 and 38) and provide the FAS map at 2 Hz below in Fig. 4. The selection of 10 Hz in the main text was based on its effectiveness across our CGM-GM model and all baseline models, ensuring a consistent and meaningful comparison. We have included additional FAS maps at 2 Hz in Figure 14 of the revised manuscript (Page 38).

FIG. 4. Comparative results of FAS maps at a frequency of 2 Hz between our generations and the empirical GMMs. This seismic event is defined with a magnitude of 3.84 and a depth of 7.94 km. The epicenter (red star) is located at a geographic position with a latitude of $37^{\circ}51.6'N$ and a longitude of $122^{\circ}15.6'W$. **a**, **b**, and **c** show the FAS maps of our generative model, ergodic GMM, and non-ergodic GMM..

Comment 7: The selection of baseline models for validation in this study appears to be inadequate. Fine-tuned GANO models such as the recently published version of [32] should be included in the comparison.

Reply: Thank you for this suggestion. We have carefully considered the fine-tuned GANO model and conducted experiments to compare its performance with our proposed CGM-GM framework. We generally followed the default network setup in the GitHub repository (<https://github.com/yzshi5/GM-GANO>) and modified the number of conditional variables from 4 (original setup in GANO) to 6 (ours). We trained the network for 500 epochs with a learning rate of 0.0001. Several other important parameters are listed here: n_{critic} is 10, λ_{grad} is 10, and the batch size is set as 64.

However, in our case, where the dataset consists of small-magnitude earthquakes with a relatively low signal-to-noise (S/N) ratio, we found that the GANO model did not perform well. Specifically, the generated waveforms exhibited noticeable discrepancies from the ground truth, as shown in both visual inspections of sample waveforms and quantitative statistical analyses of peak ground velocities and arrival times. The detailed results are shown in Fig. 5. These findings suggest that the model may struggle to generalize effectively under conditions of low S/N ratio. While we acknowledge the advancements of fine-tuned GANO models, our results indicate that their effectiveness may depend on the characteristics of the dataset, particularly for smaller-magnitude events. We appreciate the reviewer’s suggestion and welcome further discussion on this matter.

Comment 8: The model that is trained on small magnitude data, should also show the capability for reasonable scaling, with increased uncertainty, for larger magnitude events. If it doesn’t, then there is no value in having a model like this for California.

FIG. 5. Model performance of GANO on our small-magnitude earthquake datasets. **a** shows the two randomly generated samples (blue) compared to the corresponding ground truth waveforms (red). **b** and **c** present the comparisons between ground truth and the generations of peak ground velocities and arrival times, respectively.

Reply: To address your concern, we show the magnitude scaling performance of our CGM-GM model compared to empirical GMMs. Since our model is only trained on small-magnitude earthquakes ($M < 4$), directly generalizing to large-magnitude scenarios will be challenging. Therefore, to mitigate the challenge, we fine-tune our pre-trained model on two larger-magnitude earthquake data: (i) the 2014 Napa earthquake with a magnitude of 6.02 and depth of 11.0 km, and (ii) the 2018 Berkeley earthquake with a magnitude of 4.38 and depth of 12.3 km.

We present the scaling performance up to a magnitude of 8, occurring at a hypocenter depth of 7.94 km, with geographic coordinates of latitude $37^{\circ}51.6'N$ and longitude $122^{\circ}15.6'W$ (the same location as shown in Figure 2 of the manuscript). The magnitude is varied from 1 to 8, and waveforms are generated at 73 real-world stations. In Fig. 6, we present a comparison of FAS for small-magnitude earthquakes and spectral acceleration (Sa) values for large-magnitude earthquake scenarios. For small-magnitude events, we show the mean and one standard deviation of our CGM-GM predictions alongside the non-ergodic GMM. The predictions from our model generally align with those of the non-ergodic GMM, with an increased standard deviation around M1 due to the limited number of observational samples available for training, as illustrated in Figure 6(b) of the revised manuscript (Page 18). For large magnitude scaling, we evaluate the Sa results of our model compared to the ergodic ASK14 model of the Next Generation Attenuation-West2 project [1] as there are no non-ergodic models developed specifically for the SFBA.

The ASK14 model requires site and earthquake scenario parameters that may not always be readily available. To compute distances between the fault plane and the stations, we assume finite faults are centered at a latitude of $37^{\circ}51.6'N$ and a longitude of $122^{\circ}15.6'W$, which corresponds to the

hypocenter location used in small earthquake comparisons. We determine fault widths and lengths based on [3]. For site parameters, we utilize V_{s30} values ranging from 400 to 1000 m/s and $Z_{1.0}$ values between 0.1 and 2 km [2]. We then calculate Sa predictions at 10 Hz for these site conditions and take the mean values. Stations within rupture distance R_{rup} of 10 to 50 km are selected for this analysis. In Figure 6, we present comparisons for rupture distances ranging from 10 to 50 km. We plot the mean and standard deviation of the generated samples. For the GMM predictions, we calculate the logarithmic mean (μ_{mean}) of ground motion estimates, along with the mean of the standard deviation predictions.

The CGM-GM predictions exhibit an increase in Sa with increasing magnitude but remain lower than those predicted by the ASK14 model for magnitudes between 5 and 7. This discrepancy may arise because our model is trained mostly on small-magnitude events, where the earthquake source rupture processes can be approximated as point sources. In contrast, large-magnitude events require consideration of finite rupture planes, which introduces deviations in the predictions. As shown in Fig. 7, the fine-tuning strategy improves the model performance by enhancing Sa values, which further supports this observation. Furthermore, the range of ground motion predicted by the CGM-GM model is slightly narrower than that of the ASK14 model, as the generated samples represent a single realization of an earthquake scenario. These results suggest that a comprehensive investigation into the variability of the generated ground motions is an essential direction for future research. To enhance model performance, we plan to explore fine-tuning using larger datasets and incorporate out-of-distribution generalization methods. We have included additional scaling results in the revised manuscript. Please see Section *Generalization to larger-magnitude scenarios* on Pages 10-12.

Additionally, we show the comparison between the pre-trained and fine-tuned versions of our CGM-GM model. We found that the fine-tuning strategy can effectively reduce the variations of Sa values under large-magnitude scenarios. Please see Section *B.9 Details of fine-tuning and GMM comparisons* on Pages 40-41.

FIG. 6. Comparison of the magnitude scaling performance of our CGM-GM model against empirical GMMs over rupture distances between 10 and 50 km. **a** illustrates the magnitude scaling of FAS (10 Hz) for small-magnitude earthquakes (M1–4), benchmarked against the non-ergodic GMM. **b** depicts the magnitude scaling of Sa (10 Hz) for large-magnitude earthquakes (M4–8), compared with the ASK14 model.

Secondly, we respectfully disagree with the reviewer’s comment on the value of small-magnitude earthquakes in California. Accurately modeling small-magnitude earthquakes remains essential for seismic research, hazard assessment, and engineering applications in California and beyond. Small-magnitude earthquakes play a crucial role in evaluating the seismic response of infrastructures, particularly for serviceability-level assessments, performance-based earthquake engineering, and understanding long-term seismic risk in urban areas. Additionally, the study of small-magnitude earthquakes allows us to

FIG. 7. Comparison of the magnitude scaling performance between our pre-trained CGM-GM model and the fine-tuned CGM-GM model over rupture distances ranging from 10 to 50 km.

estimate the region-specific variations of the ground motion, which can lead to reducing uncertainties in the seismic hazard in the region [4,5].

We believe that developing a model for small-magnitude earthquakes in California is a meaningful and important endeavor, and we have added more clarifications on this aspect in Section *INTRODUCTION* on Page 3. The significance of simulating small-magnitude earthquakes outside California is also recognized by Reviewer 1. We cite Reviewer 1’s related comment as follows:

“In particular, the proposed method has a significant application in the improvement of shaking maps in early warning systems and seismic hazard analysis for regions with low magnitude”

References:

[1] Abrahamson, Norman A., Walter J. Silva, and Ronnie Kamai. “Summary of the ASK14 ground motion relation for active crustal regions.” *Earthquake Spectra* 30.3 (2014): 1025-1055.

[2] Hirakawa, Evan, and Brad Aagaard. “Evaluation and updates for the USGS San Francisco bay region 3d seismic velocity model in the east and North Bay portions.” *Bulletin of the Seismological Society of America* 112.4 (2022): 2070-2096.

[3] Leonard, Mark. “Earthquake fault scaling: Self-consistent relating of rupture length, width, average displacement, and moment release.” *Bulletin of the Seismological Society of America* 100.5A (2010): 1971-1988.

[4] Lavrentiadis, Grigorios, Norman A. Abrahamson, and Nicolas M. Kuehn. “A non-ergodic effective amplitude ground-motion model for California.” *Bulletin of Earthquake Engineering* 21.11 (2023): 5233-5264.

[5] Chiou, Brian, et al. “Ground-motion attenuation model for small-to-moderate shallow crustal earthquakes in California and its implications on regionalization of ground-motion prediction models.” *Earthquake spectra* 26.4 (2010): 907-926.

III. RESPONSE TO REVIEWER 3

General comment: The authors of the manuscript entitled “Learning Physics for Unveiling Hidden Earthquake Ground Motions via Conditional Generative Modeling” proposed a dynamic

VAE, featured by variational sequential architectures (e.g., RNNs) in both prior and posterior distributions, which facilitates the capture of the temporal evolution (dynamics) of time series. Thanks to this Conditional Generative Modeling for Ground Motion (CGM-GM) strategy, the authors synthesize high-frequency and spatially continuous earthquake ground motion scenarios in the San Francisco Bay Area. Interestingly, CGM-GM embeds earthquake magnitude and hypocenter coordinates as well as sensors' geographic coordinates, so to reproduce the natural ground motion variability. CGM-GM was trained with small-magnitude data, which is remarkable because they challenge small Signal-to-Noise Ratios.

The manuscript is very well written. The presented results are convincing. CGM-GM framework is effective and robust for generating earthquake ground motions. Its lightweight architecture highly appreciated. The most interesting aspect of our proposed generative model lies in its claimed ability to approximate ground motions for arbitrary (future) earthquakes and sensor locations.

Reply: Thank you for your positive feedback. We appreciate the reviewer for acknowledging the effectiveness, robustness, and lightweight design of our proposed method. We address your concerns point-by-point as follows.

Comment 1: In the Introduction, the authors overlooked some recent literature on earthquake ground motion generation using neural operators and graph neural networks. These represent a credible alternative to the CGM-GM presented in the manuscript and it would interesting to highlight what are the advantages and limitations of adopting CGM-GM instead of one of these other approaches.

Reply: In the original manuscript, our focus was on discussing the general application of generative modeling for earthquake ground motion generation. We agree with the reviewer that a more detailed introduction to other machine learning approaches, such as neural operators and graph neural networks (GNNs), would enhance the quality of the manuscript. We have expanded our discussion to include recent works on earthquake ground motion generation using neural operators [1-3] and GNNs [4-6].

Neural operators have been applied in two main paradigms for producing seismic waveforms: (i) modeling wave propagation and (ii) generative AI. While our original manuscript briefly discussed neural operators for wave propagation (see Page 2), we have incorporated additional references [2-3] highlighting recent advances in neural operators for learning 3D wave phenomena, which show promise in simulating higher-dimensional seismic events. Additionally, neural operators have been employed in a generative framework [1] for generating broadband ground motion waveforms from low-frequency components. This method considers two separate networks to learn amplitude and phase information, avoiding numerical errors from phase retrieval methods. This approach demonstrates significant improvements over traditional methods but its potential limitations lie in the high computational cost of training two independent network modules (ignoring relationships between amplitudes and phases) and the inherent instability of the GAN architecture. Additionally, GNNs have also been employed for ground motion synthesis, leveraging their ability to capture spatial correlations. These models are capable of predicting ground motion intensities at more distant stations within the seismic network.

We have included these expanded discussions and the corresponding references in the revised manuscript (see Pages 2-3). We appreciate the reviewer's feedback and welcome further recommendations if there are additional critical works we may have missed.

References:

- [1] Aquib, Tariq Anwar, and P. Martin Mai. “Broadband Ground-Motion Simulations with Machine-Learning-Based High-Frequency Waves from Fourier Neural Operators.” *Bulletin of the Seismological Society of America* 114.6 (2024): 2846-2868.
- [2] Zou, Caifeng, et al. “Deep neural Helmholtz operators for 3-D elastic wave propagation and inversion.” *Geophysical Journal International* 239.3 (2024): 1469-1484.
- [3] Lehmann, Fanny, et al. “3D elastic wave propagation with a factorized Fourier neural operator (F-FNO).” *Computer Methods in Applied Mechanics and Engineering* 420 (2024): 116718.
- [4] Clements, T., et al. “GRAPES: Earthquake early warning by passing seismic vectors through the grapevine.” *Geophysical Research Letters* 51.9 (2024): e2023GL107389.
- [5] Bloemheuvel, Stefan, et al. “Graph neural networks for multivariate time series regression with application to seismic data.” *International Journal of Data Science and Analytics* 16.3 (2023): 317-332.
- [6] Murshed, Rafid Umayer, et al. “Real-time seismic intensity prediction using self-supervised contrastive GNN for earthquake early warning.” *IEEE Transactions on Geoscience and Remote Sensing* (2024).

Comment 2: What steps and data would the authors deem necessary to apply CGM-GM to another arbitrary region of the world? The title entices the readership but it seems that CGM-GM would be hardly adoptable in regions with poor seismic record and/or varying focal mechanisms. It is in this reviewer’s opinion that the title should reflect the claim the authors make that their primary interest is in generating waveforms along the Hayward Fault, with no domain shift for the earthquake rupturing processes.

Reply: This is a thoughtful observation. The purpose of this paper is not to build a pre-trained model for Hayward fault earthquakes but to demonstrate that the conditional generative modeling framework with an appropriate parameter set can capture ground motion waveforms. We address your concerns regarding applying our method to different regions from three perspectives: requirements for implementing the CGM-GM, the results of using limited seismic records, and varying focal mechanisms. The details are outlined below. The corresponding revisions in the manuscript can be found in Section *B.8 Limited seismic records* (Pages 37-40) of Supplementary Material.

Requirements for applying the CGM-GM: The main data components include recorded seismic waveforms and their corresponding conditional variables, i.e., earthquake magnitudes, source depths, and geospatial coordinates of sources and stations. Seismic stations of interest can be selected based on a specific range of magnitudes and hypocentral distance. This model does not necessarily require V_{s30} or focal mechanisms, though they can be included as a part of conditional parameters. This feature makes the application of CGM-GM more practical, especially in scenarios where such information is either unavailable or subject to uncertainty regarding its accuracy. The implementation process can be summarized into two parts: (i) preprocessing the datasets and (ii) applying the CGM-GM framework. The preprocessing procedures are described in Section *Data selection* (Pages 15-16). We perform the Fourier transforms to both the noise and the signal and smooth the Fourier amplitudes using the Konno-Ohmachi [1] window procedure. Our goal is to retain recordings with a signal-to-noise (S/N) ratio exceeding 3 across the frequency range of [2, 15] Hz. To apply the CGM-GM model, we first extract time-frequency amplitude information using the Short-Time Fourier Transform (STFT). The dynamic VAE model is then trained on amplitude data alongside the conditional variables. Once the model is well trained, we generate synthetic ground motions by sampling from the prior distribution and conditioning on specific input variables.

Limited seismic records: To assess the robustness of our CGM-GM model under limited availability of seismic datasets, we evaluate its performance on additional scenarios with reduced station and source coverage. Specifically, we design two cases of data sparsity by independently reducing the number of stations and sources in the original training dataset. We introduce a random removal ratio λ , selected from $[0.01, 0.1, 0.2, 0.3, 0.4, 0.5, 0.9]$, to systematically reduce data availability. For each removal scenario, we train three independent models using three different random seeds, ensuring variability in the selected waveform samples and facilitating a fair comparison. For instance, applying $\lambda = 0.3$ to station removal results in a dataset containing 3880 waveforms for training. The variation in random seeds ensures different subsets of waveforms are removed in each instance. To quantify model performance, we compute the average absolute difference between the predicted and true Fourier Amplitude Spectra (FAS) values across all data samples. The results of this evaluation under reduced data availability are presented in Fig. 8. Moreover, Fig. 9 presents a comparison of FAS maps (10 Hz) for the non-ergodic GMM, CGM-GM trained with the full dataset, and CGM-GM trained under data-limited scenarios, where source removal ratios are defined as 0.3, 0.5, and 0.9. The seismic event used for producing the FAS maps is consistent with the event in Figure 2 of the manuscript, which is characterized by a magnitude of 3.84 and a depth of 7.94 km. The epicenter, denoted by a red star, is located at a geographic position with a latitude of $37^{\circ}51.6'N$ and a longitude of $122^{\circ}15.6'W$. Our model demonstrates robust performance, maintaining reasonably good performance and successfully generating realistic FAS maps even with a 30% reduction in data availability. These results validate the effectiveness of our approach in handling limited seismic records.

FIG. 8. Model performance of our CGM-GM on poor seismic records. We reduce the number of earthquake sources and stations in the original dataset. The results are based on three random runs. The purple curves represent the average absolute difference between predicted and true FAS values across all waveform samples. The shading regions denote the coverages of the mean \pm one std.

Varying focal mechanisms: In the CGM-GM framework, we use source latitudes, longitudes, and depths as conditional variables. This approach allows us to handle various focal mechanisms while implicitly assuming that focal mechanisms remain consistent (although magnitudes may vary) for earthquakes occurring at the exact same geospatial location. In our dataset, the predominant focal mechanisms are characterized by strike-slip faulting, although detailed parameters can vary [2]. Furthermore, in one of our follow-up studies, we utilized seismic records from the Geysers geothermal field [3,4]. Earthquakes in this region exhibit a range of mechanisms, including normal, reverse, and strike-slip faulting [5]. With

FIG. 9. A comparison of FAS maps at 10 Hz for the non-ergodic GMM, CGM-GM trained with the full dataset, and CGM-GM trained under data-limited scenarios, where source removal ratios are defined as 0.3, 0.5, and 0.9.

appropriate parameter settings, the CGM-GM model demonstrated the ability to effectively capture this diversity. We have included additional descriptions in Section *DISCUSSION* (Page 12).

References:

- [1] Katsuaki Konno and Tatsuo Ohmachi. "Ground-motion characteristics estimated from spectral ratio between horizontal and vertical components of microtremor." *Bulletin of the Seismological Society of America*, 88(1):228–241, 02 1998.
- [2] Hirakawa, Evan, and Brad Aagaard. "Evaluation and updates for the USGS San Francisco bay region 3d seismic velocity model in the east and North Bay portions." *Bulletin of the Seismological Society of America* 112.4 (2022): 2070–2096.
- [3] Nakata, Rie, et al. "Simulating seismic wavefields using generative artificial intelligence." *The Leading Edge* 44.2 (2025): 123–132.
- [4] Bi, Zhengfa, et al. "Advancing data-driven broadband seismic wavefield simulation with multi-conditional diffusion model." arXiv preprint arXiv:2501.14348 (2025).
- [5] Lin, Guoqing, and Bateer Wu. "Seismic velocity structure and characteristics of induced seismicity at the Geysers geothermal field, eastern California." *Geothermics* 71 (2018): 225–233.

Comment 3: Did the authors consider using ground motion numerical simulation to expand their training database? If so, how would they mix synthetics with records within the learning framework?

Reply: We have indeed considered the possibility of incorporating numerical simulation data into our training dataset. However, there are certain limitations associated with using these datasets, as they are typically constrained to low-frequency components (< 1 Hz). Furthermore, the discrepancies between simulation and observation need to be carefully examined. Therefore, we chose not to include simulation datasets in this work.

Comment 4: As per this reviewer’s understanding, each 3-component single-station time-history is encoded separately, along with its geographical coordinate. How is spatial correlation encoded into the latent space, if so? Peak values contour plots seem rather realistic, but what about the whole time series? This question arises from the fact that the phase spectra and the ground motion coherence are of paramount importance in earthquake engineering and seismic hazard modelling.

Reply: Thank you for your insightful question. In our current framework, we do not explicitly incorporate spatial correlation in the latent space. It could be achieved, e.g., by incorporating a Matérn

covariance function, as stated in Section *DISCUSSION* (see Page 12). However, spatial correlation is implicitly learned through the inclusion of geographical coordinates as conditional variables. We acknowledge that ensuring phase continuity is inherently more challenging than maintaining spatial continuity of amplitudes, due to the rapid variability of phase characteristics. Addressing this challenge is an important direction for future research, and we are actively exploring it in a separate study. We have added further clarifications in the revised manuscript. Please see Page 13.

Comment 5: The readership would benefit from a Figure showing extensive goodness-of-fit metrics comparisons. This reviewer suggests to adopt the modified Anderson’s criteria, found in Olsen et al., 2010. Other GoF metrics would probably mislead the readership, because CGM-GM is conceived for stochastic generator of ground motions, not for historical ground motion reconstruction.

Reply: Thank you for this suggestion. We have reviewed the literature and identified both Anderson’s criteria [1] and Olsen’s method (modified Anderson’s criteria) [2] as established approaches for computing goodness-of-fit (GoF) metrics for broadband waveforms. Additionally, we note that a recent generative modeling study [3] has applied Olsen’s method in this context. We acknowledge that presenting GoF metrics provides a more comprehensive evaluation of the generated waveforms and enhances the readers’ understanding of model performance. However, we were unable to find open-source implementations for directly computing GoF metrics using Olsen’s criteria. Given this, we proceeded with the original Anderson’s criteria, for which we had existing implementation resources.

Anderson’s criteria have ten distinct characteristics, each evaluated on a scale from 0 to 10. A score of 10 indicates perfect agreement. The scores for individual parameters are averaged to determine the overall GoF. A score below 4 denotes a poor fit, 4–6 represents a fair fit, 6–8 indicates a good fit, and scores exceeding 8 are an excellent fit. The GoF analysis has been performed across different rupture distances, earthquake depths, and magnitudes, with results presented in Fig. 10. Our findings indicate that the generated waveforms exhibit a good agreement with the ground truth, with mean GoF values generally around 6. We have included further clarifications in Section *B.7 Evaluations of goodness-of-fit* of the revised manuscript (Pages 36-37). Additionally, if the reviewer could provide a reference to an open-source implementation of the modified Anderson’s method, we would be happy to further evaluate it and incorporate it into our analysis.

FIG. 10. An overview of GoF values between our generations and ground truth observations across various magnitudes, distances, and earthquake depths. The dashed curves and shading regions denote the mean values and the coverages of the mean \pm one std, respectively.

References:

[1] Anderson, John G. "Quantitative measure of the goodness-of-fit of synthetic seismograms." *Proceedings of the 13th world conference on earthquake engineering*. Vol. 243. Earthquake Engineering Research Institute, 2004.

[2] Olsen, Kim B., and John E. Mayhew. "Goodness-of-fit criteria for broadband synthetic seismograms, with application to the 2008 Mw 5.4 Chino Hills, California, earthquake." *Seismological Research Letters* 81.5 (2010): 715-723.

[3] Aquib, Tariq Anwar, and P. Martin Mai. "Broadband Ground-Motion Simulations with Machine-Learning-Based High-Frequency Waves from Fourier Neural Operators." *Bulletin of the Seismological Society of America* 114.6 (2024): 2846-2868.

Comment 6: Minor comments and questions in the attached pdf.

On Page 7, "Hence, the contours of this FAS map are circular, showing geometrical spreading and average attenuation effects." Very interesting. Some similar results were found by Paolucci et al., 2018. <https://doi.org/10.1785/0120170293>. They found that the contours were following the fault geometry. Did the authors observe something similar when addressing finite-fault scenarios?

Reply: Thank you for referencing the paper by Paolucci et al. (2018) [1]. In their paper, Figures 5(a) and 5(b) illustrate heterogeneous patterns capturing physical features and "circular patterns", respectively. Their findings, including the alignment between maps of peak ground motion values and fault geometry, are highly relevant to our work.

In our manuscript, Figure 2(b) presents a radial decay pattern and average attenuation effects due to an implicit assumption of a homogeneous Earth subsurface across the spatial domain, when using earthquake magnitudes, source depths, and rupture distances as conditional variables.

Additionally, our CGM-GM predictions vary across the San Andreas and Hayward fault locations, as illustrated in Figure 2(c) of our manuscript. The sharp lithological changes lead to substantial differences in ground motions on either side of the faults. This alignment with fault geometry has also been observed in physics-based earthquake simulations of the San Francisco Bay Area [2]. This indeed align with the findings of [1] and also with the physics-based simulation results of for example [2]. Therefore, our CGM-GM model exhibits the capability to capture spatial heterogeneity in ground motions. By embedding geospatial coordinates of sources and stations as conditional variables, our model can effectively capture variations in combined site, source, and path effects, providing insights into more complex spatial features. We have expanded the discussion in the revised manuscript to include this relevant study [1,2]. Please see Section *Waveforms and spatial continuity* on Page 8.

References:

[1] Paolucci, Roberto, et al. "Broadband ground motions from 3D physics-based numerical simulations using artificial neural networks." *Bulletin of the Seismological Society of America* 108.3A (2018): 1272-1286.

[2] Rodgers, Arthur J., et al. "Broadband (0–5 Hz) fully deterministic 3D ground-motion simulations of a magnitude 7.0 Hayward fault earthquake: Comparison with empirical ground-motion models and 3D path and site effects from source normalized intensities." *Seismological Research Letters* 90.3 (2019): 1268-1284.

On Page 8, "This is due to the inductive bias of VAE models, where the learned amplitude information is over-smoothing in the logarithmic space." Have the authors tested beta-VAE in this sense?

Reply: This is a good comment. We have tried beta-VAE [1] before, and the performance was comparable to the results presented in our manuscript. While beta-VAE can help mitigate the over-smoothing issue to some extent, it is primarily designed to improve the disentanglement performance in the latent space rather than specifically address over-smoothing. The over-smoothing issue in VAEs is more directly related to the fixed variance parameter in the decoder's output distribution [2]. This fixed variance can lead to oversmoothness and posterior collapse, where the latent space becomes uninformative. Addressing this challenge requires strategies that explicitly target the decoder's variance parameterization. We have acknowledged this limitation in the manuscript and discussed potential methods for further improving model performance in section *DISCUSSION* (Page 13). For convenience, the relevant paragraph is attached below:

"Another direction is exploring specific techniques, such as generalized variance parameterization [48] and heavy-tailed distributions [49], to mitigate the over-smoothing issue in VAE models."

References:

[1] Higgins, Irina, et al. "beta-vae: Learning basic visual concepts with a constrained variational framework." *ICLR*, 2017.

[2] Takida, Yuhta, et al. "Preventing oversmoothing in VAE via generalized variance parameterization." *Neurocomputing* 509 (2022): 137-156.

On Page 11, "A potential limitation lies in the lack of the constraints of path effects." Highly appreciated comment from the authors! "The implicit learning of paths can be achievable since the training waveform data naturally contains the path effect." I suggest the work of Gatti and Clouteau 2020, <https://doi.org/10.1016/j.cma.2020.113421>.

Reply: Thank you for suggesting this interesting paper by Gatti and Clouteau (2020) [1]. Their approach is highly related to our proposed future research direction, which combines physics-based numerical simulations with experimental databases and facilitates learning the path effects. Moreover, given the practical constraints of limited data availability, we will also explore methods for explicitly incorporating path effects in the generative framework in future studies to ensure accurate modeling. We have included additional discussions on this topic in the revised manuscript and have cited the referenced paper accordingly. Please see section *DISCUSSION* on Page 13.

References:

[1] Gatti, Filippo, and Didier Clouteau. "Towards blending physics-based numerical simulations and seismic databases using generative adversarial network." *Computer Methods in Applied Mechanics and Engineering* 372 (2020): 113421.

Comment 7: Data report is not available, nor data themselves. Testing failed because of the lack of test dataset. I suggest to add some demonstrative notebook/tutorial.

Reply: Thank you for your suggestion. We have uploaded the data report to the DesignSafe [1] (see link: <https://www.designsafe-ci.org/data/browser/public/designsafe.storage.published/PRJ-4573>). The data report is also updated in the revised manuscript (see Page 20). Additionally, we have included tutorial examples, including guidance on training models and performing evaluations from various perspectives. Please see the updated materials in our GitHub repository: <https://github.com/paulpuren/cgm-gm>.

References:

[1] Lacour, M., R. Nakata, N. Nakata, N. Abrahamson (2024). Earthquake Dataset for the Study of Small Magnitude Earthquakes in the San Francisco Region. DesignSafe-CI. <https://doi.org/10.17603/ds2-necm-5q32>.

Learning Physics for Unveiling Hidden Earthquake Ground Motions via Conditional Generative Modeling

NCOMMS-24-45416A

Pu Ren, Rie Nakata, Maxime Lacour, Ilan Naiman, Nori Nakata, Jialin Song, Zhengfa Bi, Osman Asif Malik, Dmitriy Morozov, Omri Azencot, N. Benjamin Erichson, and Michael W. Mahoney

Dear Editor and Reviewers,

We sincerely appreciate the reviewers for their constructive comments and valuable suggestions. We are pleased that all reviewers acknowledge the value of our research. Our point-by-point responses are provided below marked in blue, and the corresponding manuscript revisions are highlighted in red. Notably, we have made several clarifications as follows:

- Our model is not designed to directly reproduce 3D wave propagation physics; instead, it learns a stochastic mapping that captures wave propagation characteristics between earthquake sources and stations. Please see the responses to Comments 1 and 2.
- Our model captures spatial continuity, and we have provided additional supporting evidence. Please refer to our response to Comment 3 for further details.
- We clarify that specifying only magnitudes and source locations does not uniquely define a single earthquake event. We have updated all relevant FAS maps with median values from 30 realizations. Please see Figures (2, 13-16, 23) of the revised manuscript and the responses to Comments 1 and 5.

We hope our manuscript will be found worthy of publication in this revised form.

I. RESPONSE TO REVIEWER 1

General comment: I am satisfied with the revisions to the manuscript. I recommend this article for publication.

Reply: Thank you very much for your positive feedback and recommendation for publishing.

II. RESPONSE TO REVIEWER 2

General comment: I would like to start by thanking the authors for putting effort to respond, to the extent that they have, to my questions, and improving the manuscript.

I still think that the work is valuable but its value is significantly overstated, to the point of even erroneously representing the contribution. I will summarize here, again, the main issues I have with this work. Some of the improvements that the authors have made have offered clarity while other improvements have reinforced false claims of this work. I insist that the authors should clarify the true contributions and remove all the comparisons and claims that point towards any other directions. Specifically:

Reply: We appreciate the reviewer’s continued engagement and recognition of our revisions. Your comments help us gain a deeper understanding of the strengths and remaining challenges of conditional generative models for ground motion waveform synthesis, particularly in how our model learns and represents wave propagation. Also, it is challenging to write interdisciplinary papers that make contributions both to machine learning methods and to scientific domains; and so we appreciate the feedback, which helps us to address confusion points and to clarify the contribution for both audiences. We address each of your concerns in detail below.

Comment 1: The model that the authors are presenting here is one dimensional. This means that each event-station pair is trained independently of any other station, despite the fact that the coordinates of the source and station are introduced as conditional variables. There is no global variable for the VAE to determine that two adjacent stations with the same source coordinates, from the same event, come from the same earthquake. In this sense, the model IS NOT LEARNING wave propagation.

Reply: As the reviewer points out, our model generates time series data based on earthquake source and sensor locations. We would like to clarify what physics we are trying to learn: our objective is not to simulate or learn physical wave propagation through a 3D Earth volume explicitly; instead, our model learns a stochastic mapping of a net effect of physical waveforms between source and sensor locations. We have clarified this, as we outline in detail below. By leveraging numerous earthquake-sensor pairs, our approach captures wave propagation characteristics along multiple paths through a complex neural network-based machine learning (ML) model rather than relying on a Gaussian process framework. Analogous to a non-ergodic Ground Motion Model (GMM), when our ML model provides reasonable results for a broad set of earthquake-sensor pairs, it demonstrates the capability to generalize ground motion characteristics across arbitrary source-receiver configurations within the region of interest. Furthermore, by conditioning the model on continuous geographic coordinates, our approach facilitates the inference/generation of spatially continuous waveforms, as evidenced by the smooth spatial patterns observed in the Fourier Amplitude Spectrum (FAS) maps across multiple frequencies (e.g., 2 Hz, 5 Hz, and 10 Hz). Regarding your concern about spatial continuity in Comment 3, we have provided additional clarifications and results in the corresponding response. Please see more details on Pages 4-6 of the response letter.

A primary benefit of this stochastic mapping framework, compared to explicitly modeling full 3D wave propagation, is significantly reduced complexity. Learning 3D propagation effects would require substantially larger models and considerably greater computational costs. Additionally, our model avoids explicit assumptions regarding spatial correlations. Different from Gaussian Process-based non-ergodic GMMs, where spatial correlation structures are predetermined, our approach learns these spatial dependencies directly from observed data, providing increased flexibility.

To help address the reviewer’s concern, we have revised the manuscript to clarify the following points:

- We are not learning to directly reproduce 3D wave propagation physics.
- Our model learns a stochastic mapping that captures wave propagation between earthquake sources and sensors.
- The updated median FAS maps exhibit that our model produces spatially continuous patterns. Namely, despite the conditioning of individual source-receiver pairs, the generated wavefields and FAS maps demonstrate spatial continuity, indicating our model’s capability to infer spatial coherence from data.

Moreover, we agree with the reviewer that specifying only magnitudes and source locations does not uniquely define a single earthquake event, as the generated waveforms at stations can exhibit variability over the same source parameters that we used in the conditional variables. For example, this implies that sources with identical locations and magnitudes may still differ in their source mechanisms. However, our dataset mainly includes events from one fault system, where earthquake source mechanisms show minimal variability. Thus, the variations of the wavefields among such different earthquakes are small. We demonstrate that median ground motions represent spatial continuity and have incorporated further clarifications in the Abstract (Page 1) and Section *Introduction* (Pages 3-4) of the revised manuscript.

Comment 2: Since the model is not learning wave propagation, the title LEARNING PHYSICS is false. Its learning a mapping from one point to another, independent to each other, but no coherency; and since the medium from the source to the surface is fixed, the high frequencies (here 10 Hz) are learning some shallow crustal amplification functions which one could similarly capture by replacing the source-station coordinates with V_{s30} . Instead, the authors are trying to learn a 3D problem with a 1D model which, not only is impossible, but also leads to erroneously spiked strong motion records like the ones they show in Figure 7.

Reply: As stated in our response to Comment 1, the model is not designed to reproduce full 3D wave propagation physics. Instead, it learns a stochastic mapping that captures the effective wave propagation characteristics between earthquake sources and stations. Importantly, the model demonstrates its ability to capture spatial continuity, as evidenced by the generated FAS maps (see response to Comment 3 for further details). Regarding the use of the term “learning physics” in the title, we agree with the reviewer that our model does not learn the 3D wave propagation equations. We clarify that this phrase is intended to convey the model’s ability to learn a stochastic mapping for wave propagation from data within a data-driven framework. Lest other readers have the same confusion, we have revised the Abstract and Section *INTRODUCTION* to clarify this interpretation. Please refer to Pages 3-4 of the revised manuscript.

Second, as we clarify in our response to Comment 1, we train a single ML model that can generate arbitrary pairs of source and sensors. To reproduce both dynamics (e.g., FAS) and kinematics (e.g., P and S wave arrival times and durations, Figure 4) for source-sensor pairs not included in the training dataset, learning site conditions is not sufficient. For example, site condition variations cannot explain differences in arrival times at the same source-site distances. Therefore, the model also needs to learn properties of the subsurface velocity model and how seismic waves propagate through it.

Third, not all generated waveforms exhibit spikes. We present selected examples in Section *B.1 Waveforms showing moderate performance* to illustrate specific cases. VAE models are known to generate relatively smooth amplitude information, which might introduce numerical artifacts during phase retrieval. This may partially explain the occurrence of spikes. The quality of the generated waveforms could be improved by employing alternative generative frameworks, such as diffusion models in our follow-up work [1]. We have included more clarifications in Section *DISCUSSION* (Page 13) and *B.1 Waveforms showing moderate performance* (Page 32) of the revised manuscript.

References:

[1] Bi, Zhengfa, et al. “Advancing data-driven broadband seismic wavefield simulation with multi-conditional diffusion model.” arXiv preprint arXiv:2501.14348 (2025).

Comment 3: Another way to see what I am saying is to observe the incoherent pattern that emerges in the longer period range, at 2Hz for example, where ‘path’ and ‘site’ effects become

muddled. There, Figure 14 shows that adjacent stations respond independently of each other for the same ‘event’. In other words, there is no spatial continuity. The authors are just querying the model at adjacent stations, whose waveforms, however, even for the same location of source and magnitude of earthquake, will not necessarily be spatially correlated like they are originating from the same event.

Reply: Thanks for your comment. We provide a detailed clarification on the spatial continuity in this response. We address your concern from three perspectives: (i) analysis of median FAS maps at different frequencies, (ii) evaluation of semi-variograms of the predicted ground motions, and (iii) a sensitivity test using a modified earthquake scenario with a slightly perturbed source location. In our study, the spatial continuity is achieved by conditioning the model on continuous geographic coordinates.

First, we noticed that the FAS map at 2 Hz presented in the original Figure 14 exhibited some noise and incoherent spatial patterns in certain regions. As the reviewer has suggested, we also updated all relevant FAS maps in the manuscript (Figures 2, 13-16, 23) to display the median of 30 realizations, which now provides a more stable and representative summary of the model output. Specifically, we show the revised FAS maps at 2 Hz and 5 Hz in FIG. 1 and 2, which have exhibited spatial coherence.

FIG. 1. The comparison of FAS maps at 2 Hz from CGM-GM, ergodic GMM, and non-ergodic GMM. The CGM-GM result is based on the median FAS map from 30 realizations. The red star and blue triangles represent the source and stations, respectively.

Second, in addition to the visual inspection, we compute the semi-variogram [1] of the prediction as a function of separation distance (ΔS) between the predicted stations, which is another direction for measuring spatial correlations between ground motions. As shown in FIG. 3, we observe that at very short separation distances (e.g., [0,5] km), the semi-variogram value is very close to 0, meaning that the correlation of the predicted values between nearby stations is very close to 1, which indicates that there is spatial continuity in the predictions across station locations. Moreover, we provide an analytical kernel function fitted to the empirical semivariogram, expressed as

$$\gamma(\Delta S) = 1 - \exp\left(-\frac{b \cdot \Delta S^2}{2}\right). \quad (1)$$

Here, $\gamma(\cdot)$ denotes the semi-variogram function, and b is a fitting parameter in the kernel function. The estimated values of b for the 2 Hz, 5 Hz, and 10 Hz cases are 4.858×10^{-3} , 4.033×10^{-3} , and 3.86×10^{-3} , respectively. These correspond to correlation lengths of $\rho = 14.35$, 15.75 , and 16.10

FIG. 2. The comparison of FAS maps at 5 Hz from CGM-GM, ergodic GMM, and non-ergodic GMM. The CGM-GM result is based on the median FAS map from 30 realizations.

km, respectively. The correlation length defines the spatial distance beyond which the correlation between data points significantly decreases. A correlation length of approximately 15 km is reasonable, rejecting the assumption of white noise. These findings imply the presence of spatial structure and provide further quantitative evidence supporting the spatial continuity of the generated ground motions.

FIG. 3. The empirical correlations at 2 Hz (left), 5 Hz (mid), and 10 Hz (right).

To further demonstrate spatial continuity between predictions across event locations, we slightly move the coordinates of the event location and compute new FAS maps. Specifically, the original seismic event is characterized by a magnitude of 3.84 and a depth of 7.94 km. The epicenter, denoted by a red star, is located at a geographic position with a latitude of $37^{\circ}51.6'N$ and a longitude of $122^{\circ}15.6'W$. We modify the source location 1.4 km South while keeping all other physical parameters unchanged. The comparison between the new and original FAS maps at 10 Hz is shown in FIG. 4. The resulting map exhibits a good similarity to the original, further demonstrating that our CGM-GM model maintains spatial continuity in its predictions across different event locations.

We have included an additional subsection in Supplementary Material to demonstrate the learned spatial continuity. Please refer to Section *B.11 Spatial correlations* on Pages 42-43.

Reference:

[1] Jayaram, Nirmal, and Jack W. Baker. "Correlation model for spatially distributed ground-motion intensities." *Earthquake Engineering & Structural Dynamics* 38.15 (2009): 1687-1708.

FIG. 4. The predicted FAS maps (10 Hz) with new and original source locations from our CGM-GM model. The results are both based on the median FAS values from 30 realizations.

Comment 4: In that sense, the best comparison the authors can make is against the ergodic GMMs in Nor Cal. These treat each GM time series separately with some conditional variables. The non-ergodic GMMs implement within event variability terms by studying the residuals relative to the ergodic terms of all the event terms at the same time across the strong motion network using GPs. Why dont they show in Figure 2 comparison against the ergodic GMMs? This is a much more fair comparison since the non-ergodic GMM model is a higher dimensional model than the cGM-GM. And definitely statements like ‘outperforming a state-of the art non-ergodic GMM’ should be deleted since they are misinforming the audience that the model is capturing effects that it is not.

Reply: Thank you for the suggestion. We have added a comparison between the CGM-GM predictions and the ASK14 model in terms of spectral acceleration (Sa) within the 2–15 Hz frequency range for events with magnitudes greater than 3. To compute the ASK14 predictions, we extract the V_{S30} and $Z_{1.0}$ information from the USGS San Francisco Bay Region 3D Seismic Velocity Model (Version 21.1). It is worth noting that some deviations and potential adjustments to these parameters have been suggested in prior studies [1, 2]. As shown in FIG. 5, the results show that CGM-GM consistently reduces both the residuals and their standard deviation relative to the ergodic ASK14 predictions across the entire frequency range. We have included an additional section to clarify this. Please refer to Section B.9 Residuals computed for the ASK14 model on Page 41 of the revised manuscript.

Moreover, our intention was to highlight the strong potential of our approach to complementing existing methods for generating ground motions. We deleted the sentence of “...outperforming a state-of-the-art non-ergodic GMM...” and clarified that we are aiming to complement physics-based simulations and non-ergodic empirical ground motion models. This is aspirational rather than claiming that it has already done so. In line with the reviewer’s suggestion, we have revised the corresponding statement in the Abstract of the revised manuscript to clarify this point (Page 1).

References:

- [1] Pinilla-Ramos, Camilo, et al. “Performance evaluation of the USGS velocity model for the San Francisco Bay Area.” *Earthquake Spectra* 41.1 (2025): 457-494.
- [2] Lavrentiadis, Grigorios, et al. “Data-driven characterization of near-surface velocity in the San

FIG. 5. The comparison of Sa misfits between the CGM-GM predictions and the ASK14 ergodic GMM. Residuals are defined as the logarithmic difference between the model predictions and the observed ground truth values. The solid line and the shaded area denote the mean curves and the region of mean \pm std.

Francisco bay area: A stationary and spatially varying approach.” *Earthquake Spectra* (2024): 87552930251320666.

Comment 5: Despite the fact that cGM-GM uses event and station coordinates, because the stations ‘dont know’ that their motions come from the same event, it cannot be used to plot scenaria. In that sense, all the scenaria plots like Figure 2b and 2c should be replaced with some median amplitude of many realizations, to be compared with median amplitudes of ergodic GMM plots.

Reply: We agree with the reviewer that the generated ground motions from CGM-GM at a given station cannot uniquely reflect a specific earthquake instance with the same magnitude and location, due to variability in rupture processes, source mechanisms, and source-time functions. The detailed clarifications can be found in the response to Comment 1. Moreover, as suggested by the reviewer, we have updated the relevant figure to show the median FAS computed from 30 realizations, which further improves spatial coherence while mitigating the effects of stochastic variability. Please refer to Figures (2, 13-16, 23) in the revised manuscript.

Comment 6: Page 7, line 3 of the text: [... implicit assumption of a homogeneous Earth...] should be replaced with the [... implicit assumption of a radially homogeneous Earth...]. We should also note here that this is a limitation of the baseline model that the authors selected; a more fair comparison would be to select a baseline model conditional on magnitude, distance, and a site response proxy (the same they used in their empirical GMM maps) in which case they would recover most of the capabilities that they are able to replicate with their cGM-GM model, possibly with a better time series synthesis capability.

Reply: Thanks for the suggestion. We have corrected the corresponding sentence to “...implicit assumption of a radially homogeneous Earth...” (Page 7). The non-ergodic and ergodic empirical models used in this study do not incorporate site response proxies, due to the unavailability of reliable site condition information, as described in Section *Embedding conditional variables* (Page 17) and

Empirical GMMs and Equation (9) (Pages 19-20). This limitation motivated our choice of the current baseline model.

Comment 7: Top of page 8: again erroneous statement: the agreement of the non-ergodic median here is purely phenomenological and if they plot the ergodic median the agreement will likely be better. Also, they should not be plotting scenaria, but median of scenaria, since their model is also ergodic (in the sense that it is learning independent pairs of source-stations and not within event variability like non-ergodic empirical models).

Reply: We agree with the reviewer that presenting maps from a single realization may not be sufficient. We have updated all relevant FAS maps in the revised manuscript with the median across 30 realizations instead of a single scenario. This strategy has effectively reduced pixel-level variability and provided a more stable and representative illustration of the results. Additionally, in comparison with the ASK14 model discussed in our response to Comment 4, we also compute the residuals for CGM-GM using the median values from 30 realizations. The results show that CGM-GM consistently achieves lower residuals and reduced standard deviation compared to the ergodic ASK14 predictions across the full frequency range. Further details are provided in the response to Comment 4.

Comment 8: Page 10: bottom of first paragraph: Figure 4e shows bias in the arrival times, not a good correlation as the authors claim. The heat map is clearly off center.

Reply: Thanks for pointing it out. First of all, we would like to emphasize that the duration plot (Figure 4e) is more scattered than arrival times (Figures 4c and 4d), which indicates the difficulties of accurately generating and measuring seismic durations. Second, the bias primarily occurs for durations within the 0–3 second range, corresponding to smaller-magnitude earthquakes. Due to lower signal-to-noise (S/N) ratios in such cases, it is challenging to precisely measure the durations from actual seismic data since the relatively high noise level can artificially reduce the measured duration compared to the actual signal duration. Note that the generative model does not explicitly simulate noise, resulting in inherently higher S/N ratios. This potentially amplifies the discrepancy. Nonetheless, despite these limitations, the overall duration trends between observed and generated data demonstrate a good general correlation. We have included more clarifications regarding this aspect in Section *P and S arrival times and durations* in the revised manuscript (Page 10).

Comment 9: Figure 5b and description of Figure on page 10: The SA trends at 0.1sec are completely off: the empirical model saturates at M8 due possibly to nonlinear site response and the cGM-GM increases almost exponential (due to the fictitious spikes we described above). The fact that the lines cross at some point doesn't mean there is agreement.

Reply: Thank you for your comment. We would like to clarify that our intention was not to suggest that the CGM-GM performance at 0.1 sec is acceptable. Rather, we aim to illustrate the potential capability of the CGM-GM model in capturing ground motion intensity. Specifically, by "potential", we refer to the model's ability to prevent an excessive and rapid increase in predicted intensities at higher magnitudes. To avoid confusion, we have clarified this point in the revised text and emphasized the current challenge explicitly. Please see Section *Generalization to larger-magnitude scenarios* on Pages 11-12 of the revised manuscript.

Comment 10: The 2Hz and 5Hz median predictions vs. the ergodic GMM maps should be plotted along with the 10Hz. Engineers don't care about frequencies as high as 10Hz nearly as much as they care about low frequencies.

Reply: Thank you for your comment. We agree with the reviewer and have added maps of the median predictions from CGM-GM versus ergodic GMM at 2 Hz and 5 Hz. Please see Figures 13-14 on Page 38. However, we would like to emphasize that ground-motion prediction at higher frequencies is of high importance to engineers for the design of critical infrastructures such as dams or nuclear power plants, for which the natural frequency of sensitive equipment lies in the higher frequency domain [1].

Reference:

[1] Singh, Sugandha, and Abhinav Gupta. "Seismic response of electrical equipment subjected to high-frequency ground motions." *Nuclear Engineering and Design* 374 (2021): 111046.

III. RESPONSE TO REVIEWER 3

General comment: The authors satisfactorily addressed almost all of the comments raised by this reviewer in the first round, except Comment 2. The sense of the question was if CGM-GM could be adopted zero-shot. As per the explanation given, the suspect is that the CGM-GM is not fit for zero-shot. A comment on this would be highly appreciated.

The transparency of showing the goodness of fit in Fig.10 is highly appreciated.

Reply: Thank you very much for your positive feedback. Below, we provide clarification on your question regarding zero-shot generalization.

CGM-GM is trained on a dataset of small events ($M < 4$). Within that magnitude range, the model can be used naturally in a "zero-shot" manner. However, scaling the pretrained network directly to large events introduces challenges that exceed the information (e.g., wave physics) encoded in the original training set. Specifically, the earthquake source rupture processes change from point sources to finite rupture planes. Hence, we consider supervised fine-tuning for the generalization to larger events. This reason is presented in Section *Generalization to larger-magnitude scenarios* (Pages 10-12).

Additionally, we show both zero-shot generalization (pre-trained) and fine-tuning performance in Section *B.10 Details of fine-tuning and GMM comparisons*, especially Figure 21 (Pages 41-42). The comparative results have demonstrated that the fine-tuning strategy can effectively improve the mean predictions and reduce the variations of S_a values under large-magnitude scenarios.

We hope this clarifies the intended operating regime of CGM-GM and the practical procedure for extending it to larger events.

Learning Physics for Unveiling Hidden Earthquake Ground Motions via Conditional Generative Modeling

NCOMMS-24-45416B

Pu Ren, Rie Nakata, Maxime Lacour, Ilan Naiman, Nori Nakata, Jialin Song, Zhengfa Bi, Osman Asif Malik, Dmitriy Morozov, Omri Azencot, N. Benjamin Erichson, and Michael W. Mahoney

Dear Editor and Reviewer #2,

We sincerely appreciate your constructive comments and valuable suggestions. We are pleased that Reviewer #2 acknowledged the value of our research, and we have made the suggested changes accordingly. Doing interdisciplinary work (in this case, making contributions to machine learning, earthquake engineering, and seismology) is difficult, e.g., as terminology between different fields differs. We have modified our terminology, lest any readers be confused as to our claims and contributions. Our point-by-point responses are provided below marked in blue, and the corresponding manuscript revisions are highlighted in red. We hope our manuscript will be found worthy of publication in this revised form.

I. RESPONSE TO REVIEWER 2

General comment: I would like to thank the authors for addressing my comments to some extent. There still are unresolved concerns in the revised version that I need to see addressed, however. I want to make sure that the authors properly highlight the contributions of the manuscript, which are important, without misleading the earthquake engineering audience who may be interested in using the results of their work.

For this purpose, I use sections of the text to point out inconsistencies between what the authors's model is and what they say it is in the text (line numbers would have been helpful in that sense but I'll use page numbers and approximate locations):

Reply: We thank the reviewer for acknowledging the importance of our contributions. We agree that the scope, capabilities, and limitations of our models should be precisely conveyed. Each of your concerns is addressed in detail below.

Comment 1: p3 “Our results demonstrate the excellent performance of our proposed model in learning the underlying wave propagation characteristics...” There is no evidence, nor it is expected to be any evidence of the model to learn wave propagation: for that, the authors would have to show spatiotemporal evolution of the wavefield which they don't because this is not what the model is supposed to be doing. All references to how great a job the model does in learning wave propagation should be removed. They are false. The model is learning some form of frequency response median from irregularly spaced stations and that is great. Say that.

Reply: We have clarified our statement to avoid confusion about the model's capability. The original sentence claiming the model “learns the underlying wave propagation characteristics” has been replaced with:

“Our results demonstrate the performance of our proposed model in producing Fourier Amplitude Spectra (FAS) maps and their spatial heterogeneity as well as reproducing properties such as P and S arrivals, and waveform durations.” (Page 3).

We demonstrate that the model is capable of capturing P and S arrivals and duration, as shown in the cross-plots in Figure 4(c-e). In the previous manuscript, we referred to these features collectively as “wave propagation characteristics”. This wording might not be precise, and we have revised it for accuracy.

Additionally, we have revised other related statements regarding “learning wave propagation”. Specifically, on Page 3, we have changed “This approach enables the representation of wave propagation and meaningful spatial variations in ground motions” to:

“This approach presents spatial continuity in median spectral components of the generated ground motions.”

On Page 13, we have revised “The most exciting part is that our CGM-GM can capture the underlying spatial heterogeneity and physical characteristics, as evidenced by the generation of realistic FAS maps. We conduct a comparative assessment of our CGM-GM against baseline models, including the CGM-baseline and a state-of-the-art non-ergodic GMM. The results demonstrate that our method performs comparably to the most advanced non-ergodic GMM. This validates that incorporating geospatial coordinates as conditional variables effectively enables our model to learn spatial heterogeneities.” to:

“The most exciting part is that our CGM-GM can capture the spatial heterogeneity of FAS without explicit physics constraints. We conduct a comparative assessment of our CGM-GM against baseline models, including the CGM-baseline and state-of-the-art non-ergodic GMMs. The results demonstrate that our method performs comparably to the most advanced non-ergodic GMMs.”

Comment 2: p3. “Our model learns a stochastic mapping that captures wave propagation characteristics between earthquake sources and stations...” again, rephrase the misleading statement.

Reply: We have removed this sentence “Our model learns a stochastic mapping that captures wave propagation characteristics between earthquake sources and stations, governed by physical principles and Earth structures.”, to avoid any misleading implications. Please see Pages 3-4.

Comment 3: p4. “Nevertheless, we demonstrate that the model is capable of capturing spatial continuity in the median generated ground motions.” No you dont. You demonstrate that the model is capable of capturing spatial continuity in the median generated FAS.

Reply: The sentence has been revised from “Nevertheless, we demonstrate that the model is capable of capturing spatial continuity in the median generated ground motions.” to

“Nevertheless, we demonstrate that the model is capable of capturing spatial continuity in the median generated FAS.” (Page 4).

Comment 4: p6. “We utilize an existing earthquake event that occurred at 3:16 AM on October 21, 2011.” This is misleading because, as mentioned earlier, the model is not designed to capture scenaria but median frequency responses. The authors should make it explicit that the model cannot capture event level spatiotemporal variability.

Reply: We have revised the text to avoid misleading interpretations. The original sentence “We utilize an existing earthquake event that occurred at 3:16 AM on October 21, 2011.” has been changed to:

“We use the source parameters of an existing earthquake event that occurred at 3:16 AM on October 21, 2011. Our model is not intended to reproduce ground motions of this specific event; rather, the source parameter information is used as a representative case of an earthquake along the Hayward fault.” (Page 6).

Comment 5: p13. “The primary objective of our proposed method is to generate spatially continuous and physically meaningful ground motions. Our model learns a stochastic mapping that captures wave propagation characteristics through a combined representation of source, site, and path effects.” Again, rephrase the capturing of wave propagation. Call it frequency response or something that explicitly takes out the spatiotemporal continuity misinterpretation that runs across the manuscript.

Reply: We have modified the text from “The primary objective of our proposed method is to generate spatially continuous and physically meaningful ground motions. Our model learns a stochastic mapping that captures wave propagation characteristics through a combined representation of source, site, and path effects.” to:

“Our model learns to capture the properties such as spatially continuous FAS, through a combined representation of source, site, and path effects.” (Page 13).

Comment 6: The abstract needs to be rephrased accordingly.

Reply: We have revised the abstract to align with the clarified scope and contributions of our work. Specifically, we have modified the text from “We propose a novel artificial intelligence (AI) simulator, Conditional Generative Modeling for Ground Motion (CGM-GM), to synthesize high-frequency and spatially continuous earthquake ground motion waveforms. CGM-GM leverages earthquake magnitudes and geographic coordinates of earthquakes and sensors as inputs, learning a stochastic mapping that captures wave propagation characteristics governed by physical principles and Earth heterogeneities, without explicit physics constraints” to

“We propose a novel artificial intelligence (AI) waveform generator, Conditional Generative Modeling for Ground Motion (CGM-GM). CGM-GM leverages earthquake magnitudes and geographic coordinates of earthquakes and sensors as inputs, capturing spatially continuous Fourier amplitude spectra (FAS) as well as properties such as P and S arrivals, and waveform durations without explicit physics constraints.”

Comment 7: Why compare the scaling of large magnitude events to ASK14 and not to the non-ergodic GMM for California by Lavrentiadis et al (2023)? It seems to be a much more appropriate comparison given the flow of your work than switching metric and suddenly comparing to the Sa of ASK14.

Reply: We agree with the reviewer and have updated the manuscript to compare Fourier Amplitude Spectra (FAS) results with the non-ergodic GMM for California by Lavrentiadis *et al* (2021) [1], as Lavrentiadis *et al* (2023) [2] was developed for spectral acceleration. Please see the updated Figures 5 (Page 12) and 20 (Page 42) and related descriptions (Pages 10-12, 42-43) in the manuscript.

References:

[1] G. Lavrentiadis, N. A. Abrahamson, N. M. Kuehn, A non-ergodic effective amplitude ground-motion model for California, *Bulletin of Earthquake Engineering* (2021) 1–32doi:10.1007/ s10518-021-01206-w.

[2] G. Lavrentiadis, N. Abrahamson, A non-ergodic spectral acceleration ground motion model for California developed with random vibration theory, *Bulletin of Earthquake Engineering* 21 (11) (2023) 5265–5291.

Learning Physics for Unveiling Hidden Earthquake Ground Motions via Conditional Generative Modeling

NCOMMS-24-45416C

Pu Ren, Rie Nakata, Maxime Lacour, Ilan Naiman, Nori Nakata, Jialin Song, Zhengfa Bi, Osman Asif Malik, Dmitriy Morozov, Omri Azencot, N. Benjamin Erichson, and Michael W. Mahoney

Dear Editor and Reviewer #2,

Thank you very much for your constructive comments and valuable suggestions. We particularly appreciate that Reviewer 2, who has concerns with our terminology, is helping to push forward interdisciplinary research by saying that “this group of young computer scientists with fresh ideas should gain visibility in the community.” Based on the most recent feedback, it is our understanding that the remaining concerns center on (i) the terminology involving “learning physics” and (ii) the implementation of the non-ergodic GMM. To address them, we have made two main revisions as suggested:

- Updated the title to remove “learning physics”;
- Updated the abstract to clarify the model scope.

We have also made clarifications regarding NGMM implementation below. Please see our point-by-point responses marked in blue, and the corresponding manuscript revisions are highlighted in red. We hope our manuscript will be found worthy of publication in this revised form.

I. RESPONSE TO REVIEWER 2

General comment: Again, I would like to thank the authors for taking the time to revise their manuscript and bring the manuscript content closer to what their work actually is about.

In their eagerness to achieve more, however, they –again– have introduced incorrect representations of the material presented. I am baffled by this attitude, but I do believe that this group of young computer scientists with fresh ideas should gain visibility in the community which is why I continue to suggest revisions and do not recommend that the manuscript is rejected. Here are some more items for the authors to reconsider:

Reply: We thank the reviewer for their continued effort in helping us make the manuscript more scientifically grounded. First of all, we would like to clarify that we are not a group of young computer scientists: the lead author has a background in civil and environmental engineering and now works in scientific machine learning; we have two seismologists, each with well over a decade of post-dissertation research experience, on the team; one engineering seismologist who specializes in strong motion working with leading experts in the field; and the last author is a physicist turned statistician and machine learning expert, with decades of interdisciplinary experience.

Doing interdisciplinary research is challenging since it could lead to differences in terminology across fields, which may cause unintended misunderstandings. We have taken the concerns seriously and have revised the title and abstract, as well as clarified the NGMM implementations. We address each of your concerns in detail below.

Comment 1: The title and abstract need revision. The word 'physics' is misleading in the title and now the abstract is also misleading:

[Predicting high-fidelity ground motions for future earthquakes is crucial for seismic hazard assessment and infrastructure resilience. Conventional empirical simulations suffer from sparse sensor distribution and geographically localized earthquake locations, while physics-based methods are computationally intensive and require accurate representations of Earth structures and earthquake sources. → We propose a novel artificial intelligence (AI) waveform generator, Conditional Generative Modeling for Ground Motion (CGM-GM). CGM-GM leverages earthquake magnitudes and geographic coordinates of earthquakes and sensors as inputs, capturing spatially continuous Fourier amplitude spectra (FAS) as well as properties such as P and S arrivals, and waveform durations without explicit physics constraints....]

The abstract has been edited by thereby lies the problem: the authors are not proposing a waveform generator. They are proposing an FAS median mapping tool that, if fed with a white noise phase, can produce random phase waveforms, but they are not physically correct! This is not acceptable at least on my end.

Reply: Thank you for your comment. It appears that the current title suffers from the terminology challenge, as it may give the impression that we claim to be learning complete earthquake waveform propagation physics, which is not our intent. Our method learns data-driven patterns that relate to physical properties of ground motion, including FAS, P, S-wave arrivals, and durations (as we replied in the 2nd and 3rd revisions), and we are not replacing a physics-based simulation. However, we are willing to change the title to "Learning Earthquake Ground Motions via Conditional Generative Modeling". We are also open to any other suggestions that you may have.

Second, we are willing to further revise the abstract from

"...We propose a novel artificial intelligence (AI) waveform generator, Conditional Generative Modeling for Ground Motion (CGM-GM). CGM-GM leverages earthquake magnitudes and geographic coordinates of earthquakes and sensors as inputs, capturing spatially continuous Fourier amplitude spectra (FAS) as well as properties such as P and S arrivals, and waveform durations without explicit physics constraints..."

to

"...We propose a novel artificial intelligence (AI) spectrogram generator, Conditional Generative Modeling for Ground Motion (CGM-GM). CGM-GM leverages earthquake magnitudes and geographic coordinates of earthquakes and sensors as inputs, when postprocessed with phase information, capturing spatially continuous Fourier amplitude spectra (FAS) as well as properties such as P and S arrivals, and waveform durations, without explicit physics constraints..."

Respectfully, apart from FAS median information, our method can also predict properties such as P and S arrivals, and waveform durations, as shown in Figure 4 (Page 12 of our manuscript). The conditional VAE generates spectrograms, and the phase information is obtained from the corresponding predicted amplitude data using a numerical method, the Griffin-Lim algorithm, as described in Section Network Design (Pages 15-16) of our manuscript. This phase estimation method has also been used in the prior AI-enabled waveform generation workflow [1].

Reference:

[1] Esfahani, Reza DD, et al. "TFGAN: Nonstationary ground-motion simulation in the time–frequency domain using conditional generative adversarial network (CGAN) and phase retrieval methods." *Bulletin of the Seismological Society of America* 113.1 (2023): 453-467.

Comment 2: I appreciate that the authors went ahead and incorporated the LA21 NGMM for larger events in Figure 5. Where are the continuous changes in slope in the magnitude scaling coming from ? The implementation of the NGMM is clearly incorrect and I am not sure where the error is by looking at the figure.

Reply: The NGMM prediction was obtained by combining the ergodic BA19 model [2] and the non-ergodic terms obtained by LA21 [3]. We use the values of the non-ergodic terms provided by Dr. Lavrentiadis, the author and developer of the LA21, and a software written by Dr. Bayless, the author and developer of the BA19. These softwares were used and validated in multiple engineering projects led by Dr. Abrahamson, with whom Dr. Lacour works. We confirmed with Dr. Lavrentiadis that the predictions are valid and correct. We also confirmed that the continuous change does not exist for a magnitude scaling plot of a single event-station pair, and appears when we plot all 73 stations. We consider that these were caused by substantial variations in input parameters (e.g., Vs30) across the stations in the San Francisco Bay Area [4]. Therefore, this is not a problem with our implementation.

Reference:

[2] Bayless, Jeff, and Norman A. Abrahamson. "Summary of the BA18 ground-motion model for Fourier amplitude spectra for crustal earthquakes in California." *Bulletin of the Seismological Society of America* 109.5 (2019): 2088-2105.

[3] Lavrentiadis, Grigorios, Norman A. Abrahamson, and Nicolas M. Kuehn. "A non-ergodic effective amplitude ground-motion model for California." *Bulletin of Earthquake Engineering* 21.11 (2023): 5233-5264.

[4] Hirakawa, Evan, and Brad Aagaard. "Evaluation and updates for the USGS San Francisco bay region 3d seismic velocity model in the east and North Bay portions." *Bulletin of the Seismological Society of America* 112.4 (2022): 2070-2096.

Comments 3 and 4: Furthermore, the trend of the cGM-GM and the NGMM is opposite. The proposed model doesn't saturate at large magnitude events, on the contrary – it explodes, yet another piece of evidence that it doesn't really learn physics. The authors should think really carefully of what this figure represents.

The same is evident in the high frequency regime of the small events. The model is learning a lot of noise, which is why it is predicting –at best– median amplitudes up to 11Hz. Everything above that, including the trend, should be commented on in the text, since it shows that the model doesn't extrapolate well.

Reply: Because the model is trained on small earthquake data within a limited frequency range without explicitly incorporating wave equations or subsurface information, it does not generalize to large-magnitude earthquakes (addressed in the 1st and 2nd revisions) nor perform perfectly on high-frequency ranges (as described in Lines 17-18 of Page 8 in the manuscript).